# The Role of Movement Representation Techniques in the Motor Learning Process: A Neurophysiological Hypothesis and a Narrative Review

**DOI:** 10.3390/brainsci10010027

**Published:** 2020-01-02

**Authors:** Ferran Cuenca-Martínez, Luis Suso-Martí, Jose Vicente León-Hernández, Roy La Touche

**Affiliations:** 1Departamento de Fisioterapia, Centro Superior de Estudios Universitarios La Salle, Universidad Autónoma de Madrid, 28023 Madrid, Spain; jv.leon@lasallecampus.es (J.V.L.-H.); roylatouche@yahoo.es (R.L.T.); 2Motion in Brains Research Group, Institute of Neuroscience and Sciences of the Movement (INCIMOV), Centro Superior de Estudios Universitarios La Salle, Universidad Autónoma de Madrid, 28023 Madrid, Spain; Luis.suso@gmail.com; 3Department of Physiotherapy, Cardenal Herrera University-CEU, CEU Universities, 46115 Valencia, Spain; 4Instituto de Neurociencia y Dolor Craneofacial (INDCRAN), 28008 Madrid, Spain; 5Instituto de Investigación Sanitaria del Hospital Universitario La Paz (IdiPAZ), 28029 Madrid, Spain

**Keywords:** movement representation, motor learning, motor imagery, action observation, neurophysiological hypotheses, mirror neuron system

## Abstract

We present a neurophysiological hypothesis for the role of motor imagery (MI) and action observation (AO) training in the motor learning process. The effects of movement representation in the brain and those of the cortical–subcortical networks related to planning, executing, adjusting, and automating real movements share a similar neurophysiological activity. Coupled with the influence of certain variables related to the movement representation process, this neurophysiological activity is a key component of the present hypothesis. These variables can be classified into four domains: physical, cognitive–evaluative, motivational–emotional, and direct-modulation. The neurophysiological activity underlying the creation and consolidation of mnemonic representations of motor gestures as a prerequisite to motor learning might differ between AO and MI. Together with variations in cognitive loads, these differences might explain the differing results in motor learning. The mirror neuron system appears to function more efficiently through AO training than MI, and AO is less demanding in terms of cognitive load than MI. AO might be less susceptible to the influence of variables related to movement representation.

## 1. Introduction

Movement representation training represents a revolution in the field of cognitive neuroscience and in experimental and sports psychology owing to its potential in various fields of study [1,2]. Motor imagery (MI) and action observation training (AO) are two of the most widely studied movement representation techniques. MI is defined as a cognitive and dynamic ability involving the cerebral representation of an action, without its real motor execution [3]. AO training is considered as the internal representation of a set of movements evoked by the observer during live visualization of the movements [4]. 

These movement representation techniques (both in isolation [5,6], in conjunction with various movement representation modalities [7], and in combination with real practice [8,9]) can lead to acquisition of motor gestures. It is important to evaluate what happens when movement representation techniques are applied to motor gestures and to offer a set of arguments as to why this happens. Advances in neuroimaging studies have helped answer some of the most pressing questions.

In this regard, Grush in 2004 [10] proposed one of the most relevant theories in this field, the emulation theory of representation. This theory tries to establish a theoretical framework in which, during IM, the brain constructs a visual model between the body and the environment. Subsequently, these models produce or direct an efferent sensorimotor copy in order to provide expectations or predictions of sensory feedback. These models can also be run later to create new motor images, predict results of different actions, or build new motor plans. This is the reason that visual perception is the result of using this type of model to create expectations and interpret sensory contributions during MI. In this sense, AO could provide that visual input between the subject’s body and the environment, which could facilitate the process of constructing the mental image [10].

On the other hand, Glover & Baran [11] have developed the motor–cognitive model of the MI. This model argues that central executive functions play a fundamental role during IM, but not so much in open actions. In this model, it is shown that the creation of motor mental images involves both a planning phase and a movement execution phase. To begin the creation of the mental image for the preparation of movement, an initial mental image is generated based on the motor representations stored in the nervous system. During MI and real execution, neurologically, the processes are very similar, but nevertheless, during the execution of the mental task and the execution of the real task, the processes change remarkably. During real movement, the nervous system unconsciously accesses processes of visual and proprioceptive feedback to refine the movement simultaneously with its execution. However, during MI, the control of movement creation is consciously dependent on the initial image created. That is why the ability to create motor mental images depends on the fidelity in which the subject can create the initial image. The widely developed motor actions are going to suppose a lower cognitive demand and a greater reliability in the representation, and on the contrary, the poor developed actions could create an unreliable and unprecise motor images [11].

However, despite variations in nomenclature, there are at least three established and widely described phases in the process of acquiring new motor gestures [12]. The first phase is the cognitive, characterized by the presentation of a novel gesture and the process of cognitive capture, wherein relevant information is gathered to form strategies to respond to the new demands. This phase includes an information gathering stage and a configuration of movement representation (i.e., the image of the motor gesture is constructed) [12]. 

The next two phases are the associative and automatic [12], where the motor gesture is practiced in sequences as simple as possible, until the gesture has been integrated and automated. The cognitive load is gradually reduced [13] by the action of subcortical neurophysiological structures, ultimately enabling the motor gesture to be simultaneously performed with other movements. In addition to this, it is important to stress that the repertory of motor gestures can be learned through an exploratory process. Above all, novel motor gestures. The feedback mechanism can help the learning process, as, for example, having knowledge of mistakes can consolidate the improved acquisition of a given motor gesture. Several authors have investigated the importance of feedback in motor learning process [14,15,16].

There are similarities and differences between physical practice and movement representation techniques. Therefore, the main objective of this hypothesis was to present a set of neurophysiological aspects that are likely to be involved in the motor learning process and that are mediated by MI and AO training. The secondary objective was to formulate a hypothesis to explain the differences in the effects on motor learning between AO and MI.

## 2. Effectiveness of AO and MI in the Motor Learning Process: A Minireview

Prior to the formulation of this hypothesis, a literature search was conducted to analyze whether MI and AO were effective in the process of acquiring new motor gestures. It is therefore that a minireview was carried out, which had as its main objective to see if both techniques of motion representation work in the process of motor learning.

Regarding the search strategy, the search for scientific articles was performed using PubMed (2014 to December 2019, 16th). The specific search strategy used for the database is shown below: ((((((((((((“motor”[All Fields] OR “motor’s”[All Fields]) OR “motoric”[All Fields]) OR “motorically”[All Fields]) OR “motorics”[All Fields]) OR “motoring”[All Fields]) OR “motorisation”[All Fields]) OR “motorised”[All Fields]) OR “motorization”[All Fields]) OR “motorized”[All Fields]) OR “motors”[All Fields]) AND ((((“imageries”[All Fields] OR “imagery psychotherapy”[MeSH Terms]) OR (“imagery”[All Fields] AND “psychotherapy”[All Fields])) OR “imagery psychotherapy”[All Fields]) OR “imagery”[All Fields])) OR (((“action”[All Fields] OR “action’s”[All Fields]) OR “actions”[All Fields]) AND (((((((((((((((“observability”[All Fields] OR “observable”[All Fields]) OR “observables”[All Fields]) OR “observation”[MeSH Terms]) OR “observation”[All Fields]) OR “observe”[All Fields]) OR “observed”[All Fields]) OR “observer”[All Fields]) OR “observer’s”[All Fields]) OR “observers”[All Fields]) OR “observes”[All Fields]) OR “observing”[All Fields]) OR “watchful waiting”[MeSH Terms]) OR (“watchful”[All Fields] AND “waiting”[All Fields])) OR “watchful waiting”[All Fields]) OR “observations”[All Fields]))) AND (((((((((((“motor”[All Fields] OR “motor’s”[All Fields]) OR “motoric”[All Fields]) OR “motorically”[All Fields]) OR “motorics”[All Fields]) OR “motoring”[All Fields]) OR “motorisation”[All Fields]) OR “motorised”[All Fields]) OR “motorization”[All Fields]) OR “motorized”[All Fields]) OR “motors”[All Fields]) AND ((((((“learning”[MeSH Terms] OR “learning”[All Fields]) OR “learn”[All Fields]) OR “learned”[All Fields]) OR “learning’s”[All Fields]) OR “learnings”[All Fields]) OR “learns”[All Fields])). (Filters: Randomized Controlled Trials, from 2014–2019). 

With respect to the inclusion criteria, the selection criteria used in this review were based on methodological and clinical factors, such as the population, intervention, control, outcomes, and study design (PICOS) [17] criteria as follows: 

Population: both healthy subjects and patients with any type of clinical entity susceptible to motor learning. Intervention and control: the intervention must contain at least one of the two movement representation techniques (MI or AO) in isolation or in combination with physical practice. For comparison, any other intervention different from the movement representation techniques or physical practice in isolation. Outcomes: any variable with the objective of evaluating the learning or re-learning of motor gestures. Finally, the study design: randomized controlled trials were selected. Only studies published in the last five years were considered. 

The assessment of the methodological quality of the studies was performed using the PEDro list [18]. The PEDro scale assesses the internal and external validity of a study and consists of 11 criteria: (1) specified study eligibility criteria, (2) random allocation of subjects, (3) concealed allocation, (4) measure of similarity between groups at baseline, (5) subject blinding, (6) therapist blinding, (7) assessor blinding, (8) fewer than 15% dropouts, (9) intention-to-treat analysis, (10) between-group statistical comparisons, and (11) point measures and variability data. Criteria (2)–(11) were used to calculate the PEDro score. The methodological criteria were scored as follows: yes (one point), no (zero points), or do not know (zero points). The PEDro score of each selected study provided an indicator of the methodological quality (9–10 = excellent; 6–8 = good; 4–5 = fair [18].

Two independent reviewers examined the quality of the studies selected using the same methods, and disagreements between reviewers were resolved by consensus including a third reviewer. The inter-rater reliability was determined using the Kappa coefficient, where >0.7 indicated a high level of agreement between assessors, between 0.5 and 0.7 indicated a moderate level of agreement, and <0.5 indicated a low level of agreement [19].

Regarding the results, Table 1 summarizes the results of the included studies. The total number of articles found was 21. Five studies addressed patients and 16 healthy subjects. Table 2 summarizes the methodological quality. The inter-rater reliability of the methodological quality assessment was high (*k* = 0.755). With respect to the studies included, the average score was 5.1 ± 1.54. Six studies showed good quality and 15 showed fair quality. 

To conclude this part of the manuscript, the studies of this minireview showed that movement representation techniques in both patients and healthy subjects improve the results of physical practice in isolation. Therefore, for the process of acquiring new motor gestures, physical practice should be combined with movement representation techniques to obtain better results.

## 3. Hypothesis

On the basis of the available body of evidence, we formulated a neurophysiological hypothesis regarding the potential role of movement representation techniques in the motor learning process. The components of this hypothesis are presented below.

### 3.1. Shared Neurophysiological Activity 

There is congruence between the activity of the functional neuroanatomical networks of the cortical and subcortical areas related to the planning, execution, adjustment, and automation of real movement practice and the activity that occurs during mental movement representation. This process appears to be mediated by a common neural substrate.

### 3.2. Magnitude of Brain Activity

Greater neurophysiological congruence in sensorimotor networks results in greater learning than when lesser neurophysiological congruence has occurred. A greater magnitude of neurophysiological activity, produced through movement representation, would thus lead to greater motor learning compared with lower magnitude brain activity.

### 3.3. Influence of Variables Related to Movement Representation

The magnitude of the neurophysiological activation of cortical–subcortical sensorimotor networks related to movement planning, adjustment, and execution might be modulated by the influence of certain key variables. Our hypothesis is that MI is more susceptible than AO to the influence of these key variables, owing to the inherent characteristics of the motor image construction process.

In our hypothesis, there are four domains into which we can classify these key variables: the physical domain, the cognitive–evaluator domain, the motivational–emotional domain, and the direct modulation domain of the motor representation. Table 3 summarizes the main characteristics of these variables and their estimated effect on AO and MI. 

We also propose a categorization system related to the influence of these variables on the process of movement representation. The primary variables are the direct modulation factors because they act directly on the process of live movement representation. Cognitive and physical variables could influence the direct modulation variables and the motor learning process. For example, physical activity levels could increase to generate more experience and thereby facilitate the generation of motor images. This process would also improve the understanding of the motor gesture, thereby facilitating the ability to perform the mental representation of movement. Motivational–emotional variables could influence all of these variables at all steps in the process. The visual information can help the creation of the motor representation and the set of direct modulation variables, as it can facilitate this process. This has been demonstrated in multiple studies [40,41,42,43]. The creation of the motor representation provokes a neurophysiological activation qualitatively similar to that occurring during physical practice. This has even been shown with neurovegetative activity [44]. The result of this process is the generation of mnemonic representations of movements as a prerequisite to motor learning. Figure 1 graphically represents this categorization system.

Several studies support the presence of these variables related to movement representation techniques. For example, regarding the cognitive variables, greater mental efforts made during imagery tasks led to greater hemodynamic changes at the cortical level [45]. Regarding the physical domain, there is extensive literature that supports their influence on the process of movement representation. For example, athletes with high levels of physical activity had a greater ability to generate motor images than amateur athletes with lower levels of physical activity [46,47,48]. The study conducted by La Touche et al. 2018 [49] showed that patients with chronic low back pain presented a negative correlation between the level of kinesiophobia and the ability to generate both kinesthetic and visual motor images. In addition, they also found that the ability to generate motor images was impaired in patients with chronic low-back pain compared with healthy participants. This also was found by another research group [50].

With respect to the direct modulation variables, providing visual input prior to performing an imagery motor task facilitates it and causes greater neurophysiological activity than if performed alone [42,43,51]. In addition, it has been found that the vividness of the imagination affected motor learning, showing more significant changes in those participants who presented a more vivid imagination [52]. Regarding the autonomic nervous system response, Cuenca-Martínez et al. [53] found that the complexity of movement, the effort-intensity, and the levels of physical activity can influence neurovegetative activity in the process of generating motor images. Finally, regarding the synchronization, several studies have showed that unknown, uncommon, and uncomfortable movements can lead to differences between the time employed between the imagined and real execution [54,55].

### 3.4. Differences in the Process of Creating Mnemonic Representations: Integration of Visual Information and Formation of Motor Memory

The cortical–subcortical neurophysiological activation that occurs during the representation of movements is likely to elicit the formation of specific and lasting memory imprints of the representations of the movements in the motor learning phases. Our hypothesis includes the following set of arguments regarding the creation of motor memory and the process of integrating visual information. 

The first of these arguments is that the neurophysiological paths followed by the two movement representation tools (AO and MI) during the process of acquiring and integrating visual information differ. Therefore, different strategies are employed in the process of creating the motor print. The first argument introduces the second.

The second argument is that image construction through MI is likely fed initially by the continuous activity of the working memory, and then through the activity of the episodic buffer. Figure 2 shows how this operative memory activity acts in order to integrate the visual information feeding the image construction. However, Figure 2 also shows that image construction will also receive information from episodic memory. Episodic memory feeds and is fed by semantic memory and, in the same way, by perceptual memory. Therefore, MI requires predominantly conscious strategies for the image creation process, and thus a high cognitive load, which could explain the fatigue experienced during the image construction process through MI. However, it is important to stress that it is also possible to generate images relatively unconsciously on some occasions, such as during reading. However, MI predominantly needs conscious strategies.

The third argument is that AO is not necessarily dependent on the use of conscious strategies owing to the efficiency of externally provided images. In AO, the main task is to retain and understand the image rather than create it, facilitating the working memory tasks, and thus the construction of the motor print. As a result, image transformation and a conscious effort can occur during AO, but likely require less effort than for MI.

The fourth argument is that this neurophysiological activity is optimized between the central executive control (which is part of the working memory) and procedural memory, thereby enabling the acquisition of strategies, while being unaware of the processes that govern the acquisition of those strategies. Thus, during the process of creating the motor print through AO, there is likely to be greater involvement of implicit learning with the participation of the perceptive-motor procedural memory. 

The fifth and last argument is that this activity could also respond to differences between AO and MI in susceptibility to the influence of physical, cognitive, motivational–emotional, and direct modulation variables, showing greater robustness for the influence of AO training (Figure 2).

### 3.5. Observing and Imagining: Different Cognitive Demands 

The difference between MI and AO is that all participants have the same afferent visual information arriving for processing in AO, while in MI, even though everyone receives the same verbal instructions, it is likely that there are likely to be interindividual variations that could modulate the potential of MI, and consequently the effect of MI on learning. The success of MI depends mainly on each individual’s ability to create motor images. It will also depend on the set of variables previously mentioned with the system of integration of somatosensory information, motivation, and levels of physical activity, among others.

Our hypothesis, therefore, is that the efficiency of the mirror neuron system is greater during AO training because the images are externally provided, whereas MI requires an internal, autonomous effort to create the images. This has been explicitly reported by Gatti et al. [56].

## 4. Theoretical Framework

On the basis of Finke’s functional equivalence hypothesis [57], both forms of movement representation techniques lead to the activation of areas related to the planning, generation, and adjustment of voluntary movement at the neurophysiological level. These areas include the premotor cortex, supplementary motor cortex, primary motor cortex, primary somatosensory cortex, prefrontal cortex, posterior parietal cortex, thalamus, cerebellum, and basal ganglia. The areas are activated in a similar manner to when the action is physically performed. The actions of imagining, observing, and executing an action thus converge in similar motor representations [4,58,59,60].

This overlapping functional neuroanatomy between physical practice and motion representation is also similar in terms of the magnitude and volume of brain activation [61]. However, it has been reported that this activation is lower during movement representation than during physical practice [61], a finding also reported by Lacourse et al. 2005 [60], who suggested that these differences in neurophysiological activation could be the result of striatum overactivation during the movement representation process. An inhibitory mechanism of the corticospinal signal in this subcortical structure could be acting in parallel with a cortical–subcortical activation system during the process of creating the movement representation [60].

Lacourse et al. 2005 [60] also noted that one of the main differences between the physical and non-physical practice is the lack of sensorimotor feedback during movement representation, which could provoke inactivity of somatosensory processes supporting movement representation, resulting in an exclusively top-down process, thereby limiting the effectiveness of movement representation in motor learning. The term “top-down” refers to conceptually guided systems (i.e., they start from internal processes that construct and elicit a perceptual sensory output), while bottom-up processes refer to data-driven perceptual processes, where central processes function by receiving sensory data (i.e., they begin with sensory data and end with data interpretation) [62].

The time required to perform a certain action is similar to the time taken to represent that action as a motor image [63], even when contextual variations (such as placing weights on the arms) are included [64]. Studies have also found neurophysiological similarities in the neurovegetative responses to physical practice [65] and to movement representation [66,67], even with simple motor gestures [53].

The functional relationship between movement representation and neurovegetative system activation could be based on an on-demand preparation phase (both qualitative and quantitative) of the musculoskeletal system (e.g., cardiorespiratory adaptations, sweating, body temperature adaptations) for upcoming energy expenditures [44,67].

Lacourse et al. 2005 [60] found that, after the acquisition of experience during the physical practice of a motor task and the evaluation of the movement representation process, there was no greater congruence of activation of the sensorimotor networks in comparison with physical practice than when the motor task was totally new. It has been found that the amplitude of the evoked motor potentials during AO and MI correlated positively with the ability to generate motor images [68]. Martin et al. 1990 [69] suggested that the ability to create motor images could determine the effectiveness of their use. These findings suggest that the ability to create movement representations could be a fundamental and primary modulator in the representation of motor gestures. This direct modulation, along with cognitive aspects such as understanding the motor gesture, would be present during the motor representation process. The combined physical and cognitive domains could directly influence the ability to generate motor mental images and indirectly affect motor learning. Lastly, there is the transversal motivational–emotional domain, which can influence all other domains, as well as MI and AO, owing to its effect on an individual’s predisposition towards learning. 

In an earlier study, La Touche et al. 2018 [49] found that the ability to generate motor images was impaired in patients with non-specific chronic low-back pain compared with healthy participants. Pijnenburg et al. 2015 [50] found that patients with chronic low-back pain showed a greater difference in the time performing a movement and the time spent on representing that movement. In addition, La Touche et al. 2018 [49] also found positive-moderate associations between an increased ability to create motor images and increased levels of self-efficacy, and negative-moderate associations between increased disability levels and fear of movement in patients with non-specific chronic low-back pain. These findings suggest that the three domains (physical, cognitive–evaluative, and motivational–emotional) can directly influence the ability to create movement representations. For patients with chronic pain, the information regarding the physical domain (the quality of afferent sensorimotor information, physical activity levels, and physical condition) appear to influence the direct modulation domain, thereby affecting the patient’s ability to perform certain movements.

With regard to the integration of visual information and the formation of motor memory, Mattar and Gribble 2005 [70] stated that the learning of complex motor behavior is based on the acquisition of neural representations of mechanical requirements and movement parameters (coordination, strength, speed, etc.). The authors showed that acquiring neuronal representations of the properties of motor gestures through observation was a process independent from the use of conscious strategies. This conclusion was based on the implicit properties of the sensorimotor system. The authors also found that people undergoing AO training benefited from its effects even when attentional systems were engaged in a distracting task, such as arithmetic. The authors suggested that attention systems might be involved and could influence the process, but do not appear to be critical to the observation-mediated learning process. It is possible that the mathematical distraction task demanded a specific type of cognitive task, but left free other types of cognition mechanisms sufficient for creating motor strategies [70]. 

However, it has been reported that both explicit and implicit motor learning processes can occur [71]. For example, declarative knowledge can be used to create a set of rules leading to motor learning, with the ability to obtain information on the acquisition of a set of motor gestures without being aware of the processes that govern their acquisition [72]. This acquisition, with the participation of implicit or procedural memory, can occur simultaneously with practice (a process known as “online”) or without it [73]. Explicit learning is particularly involved during the cognitive phase of motor learning when cognitive demand is high (i.e., explicit learning imposes major demands on working memory [74]). Implicit learning, however, occurs in the absence of the cognitive phase, and thus does not depend on the working memory [72].

The working memory is a complex process of active storage where information is susceptible to intra-individual manipulation. The information is consciously retained in the working memory for subsequent processing to guide behaviors [75]. One of the brain structures related to the working memory in the learning of implicit motor sequences is the dorsolateral prefrontal cortex [76]. Pascual-Leone et al. 1996 [77] found that interrupting the functioning of the contralateral dorsolateral prefrontal cortex notably affects and worsens the learning of a motor sequence.

The working memory consists of four key components [78]: the central executive, the phonological loop, the episodic buffer, and the visuospatial sketchpad. The central executive regulates the attentional process and is responsible for cognitive aspects involved in the process of information discrimination, facilitation, and inhibition. The phonological loop controls aspects related to the understanding and storage of verbal information. The episodic buffer is a storage and processing system that retrieves information from consolidated long-term memory, phonological loop, visuospatial sketchpad, and perception. The visuospatial sketchpad is related to the manipulation and reorganization of images and is relevant for planning motor gestures and retaining information on actions and objects in spatial memory [78].

Pascual-Leone et al. 1996 [77] showed the importance of the prefrontal cortex in acquiring motor gestures, and thus its role in the working memory, the latter of which requires activation of temporal and occipital regions. Visual information, therefore, appears to play an important role in the functioning of working memory and, consequently, in motor learning [78,79].

One of the most important brain structures related to the motor learning process is the cerebellum, which defines the automatic sequences associated with specific gestures [80,81]. A series of automatisms is performed through cerebellar activity, resulting in the execution of a given action. These automatisms work in conjunction with areas related to voluntary movement planning (premotor area and supplementary motor area) to select the correct motor plan [81,82]. Cerebellar functions and neurophysiological communication with secondary motor areas thus appear to be essential to the motor learning process. Lacourse et al. 2004 [83] found that movement representation increased cerebellar activity during the performance of various manual tasks, along with activity in other structures, thereby confirming the cerebellum’s influence in the automation of voluntary movement. 

Several studies have found that AO training led to greater motor learning of complex gestures in the short term than did MI [29,56]. Gatti et al. 2013 [56] argued that the human mirror neuron system, which consists of ventral premotor and lower parietal areas [84], works more efficiently, accurately, and adequately through AO. This improved functioning is because of the fact that the ventral premotor cortex (a region of the mirror neuron system largely related to the planning of voluntary movement) receives information from the visual cortex. AO training can, therefore, lead to greater functional neurophysiological activation than that provoked by MI, resulting in a greater influence on learning than MI. 

At the neurophysiological level, Loporto et al. 2011 [85] found that AO can modulate the excitability of the corticospinal system (especially premotor cortex activity) by increasing the amplitude of motor evoked potentials. The authors argued that this finding could contribute to the learning of new motor gestures.

Lacourse et al. 2005 [60] found similar congruence in sensorimotor network activation in motor image generation through MI (both unpracticed and practiced motor tasks) when compared with the physical execution of the tasks. However, Vogt et al. 2007 [84] found that, during untrained AO, there was greater activation of the premotor cortex and lower parietal cortex than when the practiced actions were observed. 

Another variable that could explain the greater impact of AO on motor learning than MI is perceived fatigue. Roure et al. 1999 [86] and Guillot et al. 2004 [87] reported that movement representation through MI can cause fatigue and difficulty maintaining attention. This loss of attention might be greater in MI-based practice than in AO training. Finally, Buccino, 2014 [4] argued that MI has been shown to have intrinsic limits that AO does not exhibit. MI appears to be a more complicated tool—in terms of cognitive demand, ability, effort, and concentration—than AO.

## 5. Conclusions

Several studies seem to support a number of the arguments presented in this hypothesis. Rizzolatti et al. 1996, 2004 [88,89] reported that the mirror neuron system, which offers the neuroanatomical support for these movement representation techniques, is widely involved in the motor learning process through movement representation 

Given that mental practice lacks the physical execution of motor actions, both the quality and quantity of neurophysiological activity in the brain regions related to generating voluntary movement are important. There also appears to be a number of variables that can modulate this activity, especially in generating motor images through MI. The motivational–emotional domain would likely influence the entire system and, together with the physical and cognitive–evaluator domains, would influence motor learning.

In the direct comparison between AO and MI, AO training appears to be more efficient for creating mnemonic representations of movements as a prerequisite to learning. AO is also less demanding in terms of cognitive load, making it more robust and less susceptible to the influence of variables related to brain representation.

Despite its disadvantages, however, MI has a relevant role. Participants can create changing scenes and diverse situations through MI. However, participants’ ability to generate motor images should be evaluated before performing MI-based interventions. The participants’ physical condition and cognitive and emotional characteristics should be considered before implementing interventions that employ movement representation techniques. Finally, both sensorimotor neurotraining tools should be considered for the acquisition of new motor gestures, in combination, combined with physical practice and in isolation, depending on the context. 

## Figures and Tables

**Figure 1 brainsci-10-00027-f001:**
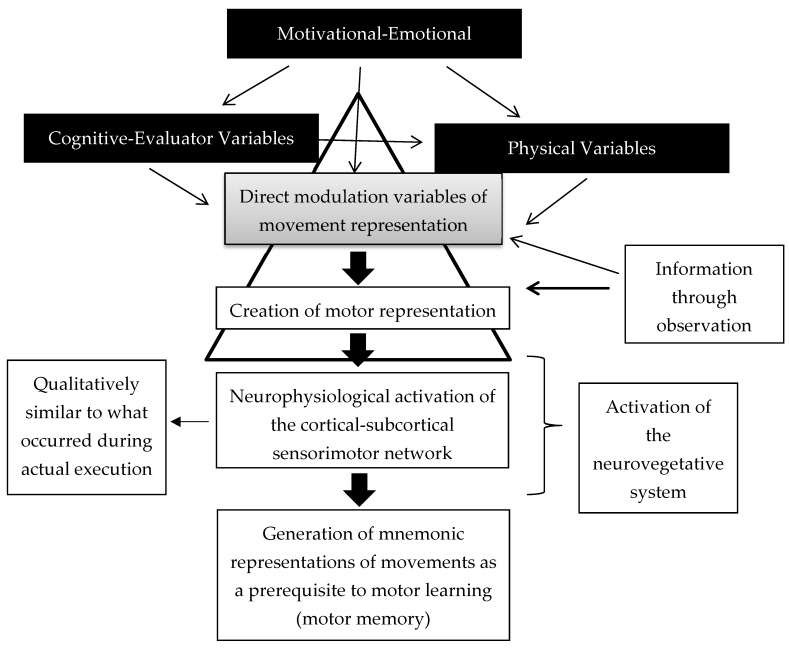
Neurophysiological view of the motor learning process mediated by movement representation techniques.

**Figure 2 brainsci-10-00027-f002:**
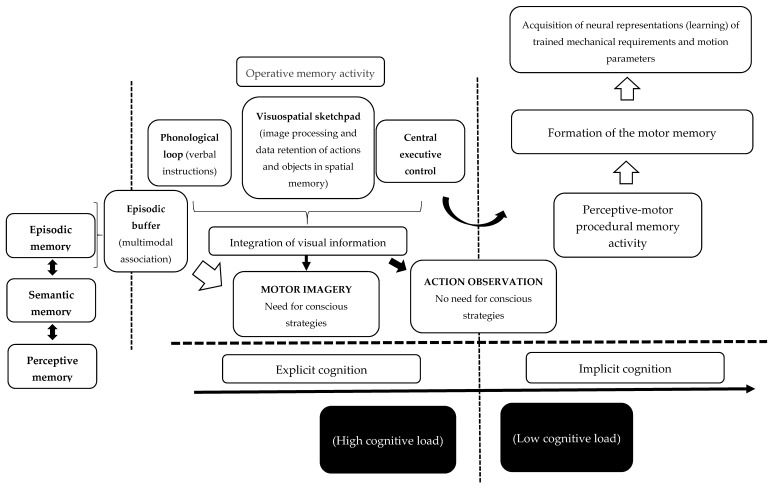
Functioning and acquisition of mnemonic representations.

**Table 1 brainsci-10-00027-t001:** Characteristics of the included studies.

Trial	Population (Patients)	Intervention Data and Target	Results
Cabral-Sequeira et al. 2016 [20]	Adolescents with cerebral palsy: 11- to 16-year-old participants (mean = 13.58 years), who suffered left (*n* = 16) or right (*n* = 15) mild hemiparesis.	EG: Day 1: MI in isolation Day 2: MI plus physical training on motor learning an aiming task CG: Day 1: recreational activities Day 2: physical training on motor learning an aiming task.	MI increased motor learning as a function of side hemiparesis in comparison with a no MI intervention.
Kumar et al. 2016 [21]	Ambulant stroke subjects: 40 hemi paretic subjects (>3 months post-stroke) who were ambulant with good imagery ability.	EG (*n* = 20): task-oriented training group plus MI CG (*n* = 20): task-oriented training group in paretic lower extremity muscles strength and gait performance.	Additional task specific MI training improves paretic muscle strength and gait performance in ambulant stroke patients.
Keynen et al. 2018 [22]	Stroke patients: 56 patients with a stroke (>3 months ago), capacity to walk independently with or without a walking aid over 10 m (with a self-selected gait speed <1.2 m/s), and presence of hemiparesis (indicated by a score of <100 on the lower extremity part of the Motricity Index and a score <34 on the lower extremity part of the Brunnstrom Fugl–Meyer assessment).	AO group *(n* = 20) Analogy instruction (*n* = 19) Environmental group (*n* = 17) To explore immediate changes in walking performance when using the three implicit learning.	Analogy instructions and environmental constraints can lead to specific, immediate changes in the walking performance and were in general experienced as feasible by the participants.
La Touche et al. 2019 [9]	Patients with chronic non-specific low back pain: (low back pain for at least the prior three months; low back pain of nonspecific nature).	MI plus physical training (*n* = 16) Tactile feedback plus physical training (*n* = 16) CG: physical training in isolation (*n* = 16). Motor control gestures acquisition.	The MI strategy was the most effective mode for developing the motor control task in an accurate and controlled manner, obtaining better outcomes than tactile feedback or verbal instruction.
Moukarzel et al. 2019 [23]	Patients with total knee arthroplasty (*n* = 24). Four men and 20 women aged from 65 to 75 years (70 ± 2.89).	EG: MI plus physical therapy program (progressive lower-extremity strengthening exercises combined with electrical stimulation for quadriceps muscle, manual therapy, knee proprioceptive exercises, gait training, and functional exercises on stairs (*n* = 12). CG: physical therapy program in isolation (*n* = 12). Quadriceps strength, peak knee flexion during the swing phase, performance at the timed up and go test, stair climbing test, six-minute walk test, and Oxford knee score.	MI showed effectiveness in gait performance and functional recovery in a small sample of patients with total knee arthroplasty.
**Trial**	**Population (Healthy Subjects)**	**Intervention Data and Target**	**Results**
Cuenca-Martínez et al. 2019 [24]	HS (*n* = 45). Fourteen men and 31 women aged from 18 to 65 years.	MI plus physical training program for the lumbo-pelvic region (*n* = 15) AO plus program for the lumbo-pelvic (*n* = 15) CG: physical training in isolation (*n* = 15). Lumbo-pelvic motor control gestures acquisition.	AO training caused faster changes in lumbo-pelvic motor control compared with the CG group. All groups showed within-group significant differences between pre- and post-intervention.
Bek et al. 2016 [25]	HS (*n* = 50). The imagery group (*n* = 18, 5 males) had a mean age of 19.4 ± .98 years, the attention group (*n* = 15, 2 males) had a mean age of 19.9 ± 1.4 years, and the control group (*n* = 17, 1 male) had a mean age of 19.8 ± 1.7 years.	Two blocks of trials were completed, and after the first block, participants were instructed to imagine performing the observed movement (imagery group, *n* = 18) or attend closely to the characteristics of the movement (attention group, *n* = 15), or received no further instructions (control group, *n* = 17). To improve imitation with imagery or attention	Both attention and motor imagery can increase the accuracy of imitation and have implications for motor learning and rehabilitation.
Sheahan et al. 2018 [26]	HS (*n* = 58). (36 females; 25.0 ± 4.1 years).	Group 1: Follow through (*n* = 8), Group 2: Planning only (*n* = 8), Group 3: MI (*n* = 16), Group 4: No motor imagery (*n* = 16), Group 5: Motor imagery no fixation (*n* = 8). Motor gestures acquisition	Results showed that simply imagining different future movements could enable the learning and expression of multiple motor skills executed over the same physical states. When subjects performed the gesture and only imagined the follow-through, substantial learning occurred.
Dana & Gozalzadeh, 2017 [27]	Young male HS (*n* = 36) (15 to 18 years).	Internal MI plus physical practice (*n* = 12) External MI plus physical practice (*n* = 12) CG: no-imagery, mental math exercise plus physical practice (*n* = 12). The performance accuracy of the groups on the serve, forehand, and backhand strokes was measured.	Results showed significant increases in the performance accuracy of all three tennis strokes in all three groups, but serve accuracy in the internal imagery group and forehand accuracy in the external imagery group showed greater improvements, while backhand accuracy was similarly improved in all three groups.
Kim et al. 2017 [28]	HS (*n* = 40), novices.	Four groups: Action observation training (*n* = 10), Motor imagery training (*n* = 10), Physical practice (*n* = 10) and no practice (*n* = 10). Golf putting performance.	Results showed that the accuracy of the putting performance were improved over time through the two types of cognitive training (AO and MI training).
Gonzalez-Rosa et al. 2014 [29]	HS (*n* = 30), non-athletes, right-handed volunteers (17 females, 13 males, mean age 22.9 + 2.3 years).	Three groups: AO watched a video of the task (*n* = 9), MI had to imagine it (*n* = 12), and CG with a distracting computation task (*n* = 9). Early learning of a complex four limb, hand-foot coordination task, and kinematic analysis.	AO showed better learning compared with MI, and also elicited a stronger activity of the sensorimotor cortex during training, resulting in a lower amount of cortical activation during task execution. During AO, subjects appear to process and collect sensory and motor information relevant to action in an effective and efficient manner, which allowed them to apply a series of decision making strategies appropriate to defining which movement sequence to perform, and activating control processes such as feed forward control during motor execution.
Hidalgo-Pérez et al. 2015 [30]	HS (*n* = 40) 24 men and 16 women aged from 18 to 65 years.	Group 1: MI plus motor control exercise (*n* = 20), Group 2: motor control exercise in isolation (*n* = 20). Sensorimotor function of the craniocervical region and the cervical kinesthetic sense.	Combining MI with the motor control exercise produced statistically significant changes in sensorimotor function variables of the craniocervical region. Cervical kinesthetic sense was not significantly different between both groups.
Ingram et al. 2016 [31]	HS (*n* = 102)	Four groups: MI or PP tested in either perceptual (altering the sensory cue) or motor (switching the hand) transfer conditions (*n* = 60). CG (*n* = 42) that did not perform a transfer condition. Perceptual and motor learning through reaction time.	Results suggested that MI-based training relies on both perceptual and motor learning, while PP-based training relied more on motor processes.
Nishizawa & Kimura, 2017 [32]	HS females (*n* = 45) mean age 20.4 + 1.7 years).	Three groups: Model- and self-observation (*n* = 15), model-observation (*n* = 15), and self-observation (*n* = 15). Motor gesture learning through the acquisition of correct sports movement.	Observation combining model and self-observation exerted a positive effect on short-term motor gesture learning.
Kawasaki et al. 2018 [33]	Elderly HS (*n* = 36) aged 60 years or older (7 women and 29 men, mean age = 70.5 ± 6.19 years).	Three groups: Unskilled or skilled model observation groups (*n* = 12, respectively), or the CG (*n* = 12). Ball rotation performance (ball rotation speed).	Results indicated that the time taken for early phase learning of a finger coordination skill was improved when an unskilled model, rather than a skilled model, was used for AO combined with MI training.
Kraeutner et al. 2016 [34]	HS (*n* = 64) right-handed participants (42 female, 22.1 ± 5.3 years).	Two groups: MI in isolation (*n* = 31) Physical practice (*n* = 33) Implicit sequence learning task.	The magnitude of the learning did not differ between groups. It is suggested that MI and physical practice are equally effective in facilitating implicit sequence learning.
Kraeutner et al. 2017 [35]	HS (*n* = 72) Right-handed subjects (49 females, 23.8 ± 7.2 years)	Four conditions of MI-based practice: 4 training blocks with a high (4-High) or low (4-Low) sequence to noise ratio, or 2 training blocks with a high (2-High) or low (2-Low) sequence to noise ratio. Implicit sequence learning task.	Results showed that the extent to which implicit sequence learning occurs through MI is impacted by manipulations to entire training time and the sequence to noise ratio. In addition, results showed that the extent of implicit sequence learning occurring through MI is a function of exposure, indicating that like physical practice, the cognitive mechanisms of MI-based implicit sequence learning rely on the formation of stimulus response associations.
Lagravinese et al. 2016 [36]	HS (*n* = 25)	(AO) training: subjects were exposed to the observation of a video showing finger tapping movements executed at 3 Hz, a frequency higher than the spontaneous one (2 Hz) for four consecutive days. The changes in motor performance and motor resonance.	Results showed that multiple sessions of AO training induced a shift of the speed of execution of finger tapping movements toward the observed one and a change in motor resonance.
Lei et al. 2016 [37]	HS (*n* = 47) right-handed individuals (23 men, 17 women), aged from 18 to 30 years.	Five conditions: - AO, in which the subjects watched a video of a model who adapted to a novel visuomotor rotation - Proprioceptive training, in which the subject’s arm was moved passively to target locations that were associated with desired trajectories - Combined training, in which the subjects watched the video of a model during a half of the session and experienced passive movements during the other half - Active training, in which the subjects adapted actively to the rotation - A control condition, in which the subjects did not perform any task.	Results showed an improvement in visuomotor adaptation following the action observation, as compared with the adaptation performed by the individuals who were naïve to the given visuomotor rotation
Salfi et al. 2019 [38]	HS (*n* = 39) (aged 24.9 ± 3.0 years; range, 20–34; 18 males).	MI and Targeted memory reactivation. Four conditions: - MI in isolation - MI with an incompatible sound stimulation - AO - Auditory targeted memory reactivation during AO To assess the influence on performance on a sequential finger tapping task of an auditory targeted memory reactivation during MI practice.	The combination of MI and targeted memory reactivation showed the largest early performance improvement, as indexed by the combined measure of speed and accuracy
Sobierajewicz et al. 2016 [39]	HS (*n* = 24) 6 males and 18 females range 21 to 28 years.	After an informative cue, a response sequence had either to be executed, imagined, or withheld. The learning of a fine hand motor gesture.	Both physical condition and MI condition improved the response time and accuracy although the effect of motor learning by motor imagery was smaller than the effect of physical practice

EG: experimental group, CG: control group, MI: motor imagery; AO: action observation; HS: healthy subjects, PP: physical practice.

**Table 2 brainsci-10-00027-t002:** Assessment of the studies quality based on the PEDro scale.

Items												
	1	2	3	4	5	6	7	8	9	10	11	Total
Cabral-Sequeira et al. 2016 [20]	1	1	0	1	0	0	0	1	0	1	1	**5**
Kumar et al. 2016 [21]	1	1	1	1	0	0	1	1	0	1	1	**7**
Keynen et al. 2018 [22]	1	1	1	1	0	0	0	1	0	0	1	**5**
La Touche et al. 2019 [9]	1	1	1	1	1	0	1	1	0	1	1	**8**
Moukarzel et al. 2019 [23]	1	1	0	1	1	0	1	1	0	1	1	**7**
Cuenca-Martínez et al. 2019 [24]	1	1	1	1	1	0	1	1	0	1	1	**8**
Bek et al. 2016 [25]	1	1	0	1	0	0	0	1	0	1	1	**5**
Sheahan et al. 2018 [26]	1	0	0	1	0	0	0	1	0	1	1	**4**
Dana & Gozalzadeh 2017 [27]	1	0	0	0	0	0	0	1	0	1	1	**3**
Kim et al. 2017 [28]	1	1	0	1	0	0	0	1	0	1	1	**5**
González-Rosa et al. 2014 [29]	1	1	0	1	0	0	0	1	0	1	1	**5**
Hidalgo-Pérez et al. 2015 [30]	1	1	1	1	0	0	1	1	0	1	1	**7**
Ingram et al. 2016 [31]	1	1	0	1	0	0	0	1	0	1	1	**5**
Nishizawa & Kimura, 2017 [32]	1	1	0	1	0	0	0	1	0	1	0	**4**
Kawasaki et al. 2018 [33]	1	1	1	1	0	0	0	1	0	1	1	**6**
Kraeutner et al. 2016 [34]	1	1	0	0	0	0	0	0	0	1	1	**4**
Kraeutner et al. 2017 [35]	1	1	0	1	0	0	0	0	0	1	1	**4**
Lagravinese et al. 2016 [36]	1	0	0	1	1	0	0	1	0	1	1	**5**
Lei et al. 2016 [37]	1	0	0	1	0	0	0	1	0	1	0	**3**
Salfi et al. 2019 [38]	1	0	0	1	0	0	0	1	0	1	1	**4**
Sobierajewicz et al. 2016 [39]	1	0	0	1	0	0	0	0	0	1	1	**3**

1: subject choice criteria are specified; 2: random assignment of subjects to groups; 3: hidden assignment; 4: groups were similar at baseline; 5: all subjects were blinded; 6: all therapists were blinded; 7: all evaluators were blinded; 8: measures of at least one of the key outcomes were obtained from more than 85% of baseline subjects; 9: intention-to-treat analysis was performed; 10: results from statistical comparisons between groups were reported for at least one key outcome; 11: the study provides point and variability measures for at least one key outcome.

**Table 3 brainsci-10-00027-t003:** Modulating variables of the movement representation process.

Domain	Variables	Influence
**Physical** **MI ***** **AO ***	- Levels of physical activity	- Greater physical activity levels might generate greater facility in constructing the movement due to the experience, development, and elaboration of habitual motor schemes.
	- Perceived of mental fatigue	- The presence of high fatigue levels can affect attention, thereby limiting the brain’s construction of movement.
	- Disturbances in sensorimotor integration	- The presence of somatosensory disturbances can generate aberrant sensorimotor schemes that could affect the movement’s construction, thereby leading to a decreased ability to generate motor images.
**Cognitive–Evaluator** **MI ***** **AO ***	- Understanding motor gestures and verbal instructions	- Understanding movements that are not physically elaborated can improve the planning phases of movement because emotional and cognitive limitations can be reduced.
	- Context	- The development of the movement in family and specific contexts could facilitate imagination and observation.
	- Functioning of the working memory	- Better functioning of the working memory could increase the ability to collect the provided information and its subsequent consolidation into long-term memory, thereby facilitating the motor learning process.
	- Self-efficacy levels	- Greater self-perception of the ability to generate motor images could enhance the brain’s ability to construct motor images.
	- Attention levels	- Maintaining attention could facilitate the mental construction of movements and the total effort dedicated to that construction.
	- Expectations	- Expectations of the effects of movement representation techniques might influence the efficiency of the motor learning process.
	- Perception of difficulty	- Greater perception of the difficulty could lead to a reduced ability to generate motor representation and thereby worsen motor learning.
**Motivational–Emotional** **MI ***** **AO *****	- Motivation (reasons, intention, and desires)	- Higher motivation levels could lead directly to a better predisposition towards the learning process and, therefore, on the effects of movement representation techniques.
	- Fear of movement	- Higher kinesiophobia levels can lead to an interruption of the motion representation process, thereby impairing the motor learning process.
**Direct modulation** **MI ***** **AO ***	- Ability to create motor images	- The effectiveness of MI might depend on the ability to create motor images. This aspect can be influenced by other domains.
	- Synchronization	- Greater time congruence between physical practice and motion representation could facilitate the motor learning process.
	- Activity of the autonomous nervous system	- Greater neurovegetative activity could indicate higher neurophysiological activity of the sensoriomotor cortical–subcortical networks, indicating greater effort dedicated to the task, greater attention, and less fatigue, thereby favoring motor learning.

* low susceptibility; ** moderate susceptibility; *** high susceptibility. Abbreviations: AO, action observation; MI, motor imagery.

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
