# Peer review of "The Role of Movement Representation Techniques in the Motor Learning Process: A Neurophysiological Hypothesis and a Narrative Review"

_brainsci, 2020, doi:10.3390/brainsci10010027_

Round 1

Reviewer 1 Report

The role of movement representation techniques in the motor learning process: A neurophysiological hypothesis and a narrative review.

Short summary

The article recapitulates recent findings in motor imagery and in action observation research. This includes neurophysiological similarities between action execution and motor imagery as well as similarities between action execution and action observation. The authors present four domains/variables/categories that are somehow related to both motor imagery and action observation.

I always sign my reviews: Stephan Dahm

Major Comments

The manuscript type is called 'Hypothesis'. I could only find 3 types of manuscript types for Brain Sciences that are Articles, Reviews, Case Reports.
As there is no data provided in the manuscript, the only option left is Reviews. However, this is not a systematic review. I would recommend to do a systematic review on similarities and differences between MI and AO.

The English writing style of the present manuscript is very well accomplished, even so the manuscript is quite hard to follow. First, terms are not introduced or not consistently used. I will give two examples, but there are much inconsistently used terms. What is the difference between brain training (line 42) and movement representation techniques (title)? What is the difference between motor skills (line 42) and motor gestures (line 43)? If you intend to express the same thing, please be concise and use the same term throughout the whole manuscript. Second, the aim of the manuscript is not clear. In my opinion the aim of the manuscript cannot be to formulate a hypothesis, but rather to test a hypothesis. A theoretical framework is provided, but it remains unclear how this framework has been developed and where it comes from. Further, it remains unclear what this framework exactly explains. Is it the idea to make up a framework that integrates both MI and AO together?

Many passages lack literature references. These are at least partially provided at later stages of the manuscript. However, they need to be included in the second part of the manuscript. I’d suggest to combine ‘2. Hypothesis’ and ‘3. Evidence’ to one part called ‘Theoretical framework’. It would add incremental value to the manuscript if the framework would be tested with data. If the framework is not tested in the present study, it should be stated how this could be done in future studies.

Again use of terms, but now the main terms. Please clearly distinguish between motor imagery and mental practice. The same holds for AO and using AO to improve motor execution. In parts of the manuscript, it remains unclear whether MI and ME differ in durations, differ in vividness, or whether the repeated use of MI and ME does not result in the same performance enhancements. Also explain what you mean with ‘movement representation training’ (line 33). It could mean learning a physical skill with movement representation techniques, but also to train to improve movement representation (more vivid in MI, better focus in AO). Further, I would recommend not to use the term ‘brain training’ (line 41 and others). I associate this with doing Sudoku grids to stay clever, but that could be my personal opinion.

In line 40 literature is cited that investigated MI in isolation, AO in isolation, both in combination with ‘various brain training modalities’, and in combination with real practice. I had the impression that you missed a whole bunch of studies of David Wright and colleagues who had been working on what they call AOMI. There has been very much work on this topic in the last five years. I would further suggest to include theory on MI and AO provided by Grush (2004) and Glover & Baran (2017). Then highlight more clearly which are the new insights of the present manuscript.

In the beginning of page 2, motor learning by physical practice is explained. I believe this could be done with more detail. ‘The presentation of a novel gesture’ (line 47) only applies for motor learning by external instruction. However, I believe that that much of our motor skills repertoire is learned by exploration. This means doing something random and then remembering it to repeat it at later stages. This is how children and adults learn a new motor skill (not a new combination of already known skills). For instance, a new motor skill could be to move your ears. I am totally in line with the authors that oral instructions, AO, or MI may fasten the learning process. However, feedback (for moving the ears visual feedback from a mirror) should also be taken into account. Further, errors as a driving mechanism in motor learning should be taken into account. There are studies investigating action consequences in MI (Dahm & Rieger, 2019). There are also studies investigating the impact of action errors on motor learning in AO by manipulating the Video material (showing actors who make errors or not, showing real participants who naturally err).

Please explain where the four categories (physical, cognitive, motivational-emotional, direct modulation) come from. Provide references and explain how these categories were selected. The last category seems not to fit to the other three categories. Further, explain why you’d expect the influences described in Table 1 and provide references. Where do the starts in Table 1 come from? Please explain and provide references. I like the idea to disentangle all the variables listed in Table 1. However, this could be done more concisely and not necessarily in a table. What do you mean with ‘influence’? 1. The influence of the variable on the ease/vividness of MI? Is there something similar to ease/vividness in AO? 2. The influence of the variable on motor learning (improvements in physical execution of the action). Maybe explain the four possible types of influence for each variable separately?

The three paragraphs (line 192-228) are 3 independent summaries of three previous studies. A red line is missing as in many parts of the manuscript. There are many interesting points raised in the manuscript, but they are not logically connected. They are rather set one after another. The same holds for the information in the figures.

Minor Comments

 I would suggest to structure the sentences in a way that the authors are not mentioned in the text. This would not only shorten the manuscript, but also make it easier to read. I will give an example: ‘MI is defined as a cognitive and dynamic ability involving the cerebral representation of an action without its real motor execution [3]’ (line 36).

Abstract line 27: ‘AO is less demanding in terms of cognitive load than MI’.

Please use ‘physical practice’ instead of ‘actual practice’. It fits better in contrast to mental practice.

Line 77: ‘the most susceptible of the movement representation techniques’. How many techniques are there? So far you present only two: MI and AO. Did I miss something?

Table 1 ‘physical activity level’: Do you mean that it is easier imagining a high jump while riding a bike?

Table 1 ‘fatigue’: perceived fatigue and objective muscle fatigue is not the same. Please be precise in the description of the influence.

Figure 1: What is the new insight that I get from this framework? Is it the four categories in black? Please explain this to the reader. Also, there are quite a lot of arrows in the figure. Please explain them in the text and provide references with data that made you conclude this.

Line 123: Is tools and techniques the same? Is there a difference?

Line 124 ‘different strategies’: Please explain them in detail.

On page 6 it is explained that AO is unconscious, but MI is conscious. This may not always be the case. For instance, if someone reads a text that includes action words, the reader may imagine the action involuntarily.

Line 145: ‘The difference between MI AO is that all participants have the same afferent visual information arriving for processing in AO …’. I agree that the video should be the same for all subjects. But similarly the instructions in MI should always be the same for all subjects. This is the objective stimulus material which is the same for all subjects in both MI and AO. What participants do with the stimulus material (instructions, videos etc.) is what is here called ‘processing’. I would say that both in MI and AO, participants may have an individual focus on certain details of the action. Please explain why this is different in MI and AO.

Line 149: ‘…efficiency of the mirror neuron system is greater during AO…’ than during MI. Please further explain what you mean with ‘efficiency’. Do the neurons fire less often with a higher outcome? What is the outcome? Please provide references for your statement.

Line 168: ‘the two practices’ which ones do you mean? MI? AO? Physical practice?

I am not sure whether figure 2 improves readability and understanding of the text. Particularly because the content is not further explained in the text.

Line 279ff: What do the presented results have to do with ‘improvements in the mirror neuron system’?

Reference 8 (line 334). ‘Le Effects of motor imagery’

Author Response

December 19, 2019

Dear Editorial Office,

We are pleased to submit our point-by-point response to the changes requested on our paper “The Role of Movement Representation Techniques in the Motor Learning Process: A Neurophysiological Hypothesis and a Systematic Review".

We believe that we have made all the changes requested prior to publication and our manuscript is now responsive to all comments and suggestions. We highlighted the changes along the text with red color to show reviewers and editor where the changes have been made. In addition, our responses to the reviewers in this document are highlighted in red color.

We would like to thank the journal Editor of Brain Sciences for having managed those different referring researchers in this area could have reviewed our manuscript. Thanks to them, the manuscript has remarkably improved its quality.

We look forward to hearing your response and thank you for your consideration in bringing this manuscript closer to publication.

Sincerely yours,

The authors

Author Response

Reviewer 1

Short summary

The article recapitulates recent findings in motor imagery and in action observation research. This includes neurophysiological similarities between action execution and motor imagery as well as similarities between action execution and action observation. The authors present four domains/variables/categories that are somehow related to both motor imagery and action observation.

I always sign my reviews: Stephan Dahm

Response: First, we would like to highlight the excellent work of Dr. Dahm because he has considerably increased the quality of this manuscript. The authors thank him for his excellent work.

Major Comments

The manuscript type is called 'Hypothesis'. I could only find 3 types of manuscript types for Brain Sciences that are Articles, Reviews, Case Reports.

As there is no data provided in the manuscript, the only option left is Reviews. However, this is not a systematic review. I would recommend to do a systematic review on similarities and differences between MI and AO.

Response: Thank you for this comment. The paper type has been changed to Review.

However, we believe that this request can be fulfilled and can give a quality leap to this manuscript. It is therefore that we have carried out a systematic search with the following search strategy to compare similarities and differences in the two techniques of movement representation. Studies in both patients and healthy subjects have been included within the last 5 years. Table 1 summarizes the studies.

(Text)

Effectiveness of AO and MI in the motor learning process: a minireview

Prior to the formulation of this hypothesis, a literature search was conducted to analyze whether MI and AO were effective in the process of acquiring new motor gestures. It is therefore that a minireview was carried out which had as its main objective to see if both techniques of motion representation work in the process of motor learning.

Regarding the search strategy, the search for scientific articles was performed using PubMed (2014 to December 2019, 16th). Specific search strategy used for the database is shown below:

(((((((((((("motor"[All Fields] OR "motor's"[All Fields]) OR "motoric"[All Fields]) OR "motorically"[All Fields]) OR "motorics"[All Fields]) OR "motoring"[All Fields]) OR "motorisation"[All Fields]) OR "motorised"[All Fields]) OR "motorization"[All Fields]) OR "motorized"[All Fields]) OR "motors"[All Fields]) AND (((("imageries"[All Fields] OR "imagery psychotherapy"[MeSH Terms]) OR ("imagery"[All Fields] AND "psychotherapy"[All Fields])) OR "imagery psychotherapy"[All Fields]) OR "imagery"[All Fields])) OR ((("action"[All Fields] OR "action's"[All Fields]) OR "actions"[All Fields]) AND ((((((((((((((("observability"[All Fields] OR "observable"[All Fields]) OR "observables"[All Fields]) OR "observation"[MeSH Terms]) OR "observation"[All Fields]) OR "observe"[All Fields]) OR "observed"[All Fields]) OR "observer"[All Fields]) OR "observer's"[All Fields]) OR "observers"[All Fields]) OR "observes"[All Fields]) OR "observing"[All Fields]) OR "watchful waiting"[MeSH Terms]) OR ("watchful"[All Fields] AND "waiting"[All Fields])) OR "watchful waiting"[All Fields]) OR "observations"[All Fields]))) AND ((((((((((("motor"[All Fields] OR "motor's"[All Fields]) OR "motoric"[All Fields]) OR "motorically"[All Fields]) OR "motorics"[All Fields]) OR "motoring"[All Fields]) OR "motorisation"[All Fields]) OR "motorised"[All Fields]) OR "motorization"[All Fields]) OR "motorized"[All Fields]) OR "motors"[All Fields]) AND (((((("learning"[MeSH Terms] OR "learning"[All Fields]) OR "learn"[All Fields]) OR "learned"[All Fields]) OR "learning's"[All Fields]) OR "learnings"[All Fields]) OR "learns"[All Fields])). (Filters: Randomized Controlled Trials, from 2014 – 2019).

With respect the inclusion criteria, the selection criteria used in this review were based on methodological and clinical factors, such as the population, intervention, control, outcomes, and study design (PICOS) [15] criteria as follows:

Population: both healthy subjects and patients with any type of clinical entity susceptible to motor learning. Intervention and control: the intervention must contain at least one of the two movement representation techniques (MI or AO) in isolation or in combination with physical practice. For comparison, any other intervention different from the movement representation techniques or physical practice in isolation. Outcomes: any variable with the objective of evaluating the learning or re-learning of motor gestures. Finally, the study design: randomized controlled trials were selected. Only studies published in the last 5 years were considered.

The assessment of the methodological quality of the studies was performed using the PEDro list [16]. The PEDro scale assesses the internal and external validity of a study and consists of 11 criteria: 1) specified study eligibility criteria, 2) random allocation of subjects, 3) concealed allocation, 4) measure of similarity between groups at baseline, 5) subject blinding, 6) therapist blinding, 7) assessor blinding, 8) fewer than 15% dropouts, 9) intention-to-treat analysis, 10) between-group statistical comparisons, and 11) point measures and variability data. Criteria (2) – (11) were used to calculate the PEDro score. The methodological criteria were scored as: Yes (one point), No (zero points) or don’t know (zero points). The PEDro score of each selected study provided an indicator of the methodological quality (9–10 = excellent; 6–8 = good; 4–5 = fair [16].

Two independent reviewers examined the quality of the studies selected using the same methods, and disagreements between reviewers were resolved by consensus including a third reviewer. The inter-rater reliability was determined using the Kappa coefficient, where > 0.7 indicated a high level of agreement between assessors, between 0.5 and 0.7 indicated a moderate level of agreement, and < 0.5 indicated a low level of agreement [17].

Regarding the results Table 1 summarizes the results of the included studies. The total number of articles found was 21. Five studies addressed patients and 16 healthy subjects. Table 2 summarizes the methodological quality. The inter-rater reliability of the methodological quality assessment was high (k = 0.755). With respect to the studies included, the average score was 5.1 ± 1.54. Six studies showed good quality and 15 fair quality.

To conclude this part of the manuscript, the studies of this minireview showed that movement representation techniques in both patients and healthy subjects improve the results of physical practice in isolation. Therefore, for the process of acquiring new motor gestures, physical practice should be combined with movement representation techniques to obtain better results.

Table 1. Characteristics of the included studies

Trial

Population (Patients)

Intervention data and target

Results

Cabral-Sequeira et al 2016

Adolescents with cerebral palsy:

11- to 16-year-old participants (mean = 13.58 years), who suffered left (n = 16) or right (n = 15) mild hemiparesis.

EG:

Day 1: MI in isolation

Day 2: MI plus physical training on motor learning an aiming task

CG:

Day 1: recreational activities

Day 2: physical training on motor learning an aiming task.

MI increased motor learning as a function of side hemiparesis in comparison with a no MI intervention.

Kumar et al. 2016

Ambulant Stroke subjects:

40 hemi paretic subjects (>3 months post-stroke) who were ambulant with good imagery ability.

EG (n = 20):

task-oriented training group plus MI

CG (n = 20):

task-oriented training group

in paretic lower extremity muscles strength and gait performance.

Additional task specific MI training improves paretic muscle strength and gait performance in ambulant stroke patients.

Keynen et al. 2018

Stroke patients:

56 patients with a stroke (>three months ago), capacity to walk independently with or without

a walking aid over 10 m (with a self-selected gait speed <1.2 m/s) and presence of hemiparesis

(Indicated by a score of <100 on the lower extremity part of the Motricity Index and a score <34 on the lower extremity part of the Brunnstrom Fugl-Meyer assessment).

AO group (n = 20)

Analogy instruction (n = 19)

Environmental group (n = 17)

To explore immediate changes in walking performance when using the three implicit learning.

Analogy instructions and environmental constraints can lead to specific, immediate changes in the walking performance and were in general experienced as feasible by the participants.

La Touche et al. 2019

Patients with chronic non-specific low back pain:

(-low back pain for at least the prior 3months; -low back pain of nonspecific nature).

MI plus physical training (n=16)

Tactile feedback plus physical training (n = 16)

CG: physical training in isolation (n = 16).

Motor control gestures acquisition.

The MI strategy was the most effective mode for developing the motor control task in an accurate and controlled manner, obtaining better outcomes than tactile feedback or verbal instruction.

Moukarzel et al. 2019

Patients with total knee arthroplasty (n = 24). 4 men and 20 women aged from 65 to 75 years (70±2.89).

EG: MI plus physical therapy program (progressive lower-extremity strengthening exercises combined with electrical stimulation for quadriceps muscle, manual therapy, knee proprioceptive exercises, gait training and functional exercises on stairs (n = 12).

CG: physical therapy program in isolation (n = 12).

Quadriceps strength, peak knee flexion during the swing phase, performance at the timed up and go test, stair climbing test, 6-minute walk test and Oxford knee score.

MI showed effectiveness in gait performance and functional recovery in a small sample of patients with total knee arthroplasty.

Trial

Population (Healthy subjects)

Intervention data and target

Results

Cuenca-Martínez et al. 2019

HS (n = 45). 14 men and 31 women aged from 18 to 65 years.

MI plus physical training program for the lumbo-pelvic region (n=15)

AO plus program for the lumbo-pelvic (n = 15)

CG: physical training in isolation (n = 15).

Lumbo-pelvic motor control gestures acquisition.

AO training caused faster changes in lumbo-pelvic motor control compared with the CG group. All groups showed within-group significant differences between pre- and post-intervention.

Bek et al. 2016

HS (n = 50).

The Imagery group (n = 18, 5 males) had a mean age of

19.4 ± .98 years, the Attention group (n = 15, 2 males)

had a mean age of 19.9 ± 1.4 years, and the Control group

(n = 17, 1 male) had a mean age of 19.8 ± 1.7 years.

Two blocks of trials were completed, and after the first block participants were instructed to imagine performing the observed movement (Imagery group, n = 18) or attend closely to the characteristics of the movement (Attention group, n = 15), or received no further instructions (Control group, n = 17).

To improve imitation with imagery or attention

Both attention and motor imagery can increase the accuracy of imitation and have implications for motor learning and rehabilitation.

Sheahan et al. 2018

HS (n = 58). (36 females; 25.0 ± 4.1 years).

Group 1: Follow through (n = 8),

Group 2: Planning only (n = 8),

Group 3: MI (n = 16),

Group 4: No motor imagery (n = 16), Group 5: Motor imagery no fixation (n = 8).

Motor gestures acquisition

Results showed that simply imagining different future movements could enable the learning and expression of multiple motor skills executed over the same physical states.

When subjects performed the gesture and only imagined the follow-through, substantial learning occurred.

Dana & Gozalzadeh, 2017

Young male HS (n = 36). (15 to 18 years).

Internal MI plus physical practice (n=12)

External MI plus physical practice (n = 12)

CG: no-imagery, mental math exercise plus physical practice (n = 12).

The performance accuracy of the groups on the serve, forehand, and backhand strokes was measured.

Results showed significant increases in the performance accuracy of all three tennis strokes in all three groups, but serve accuracy in the internal imagery group and forehand accuracy in the external imagery group showed greater improvements, while backhand accuracy was similarly improved in all three groups.

Kim et al. 2017

HS (n = 40), novices.

Four groups:

Action observation training (n = 10),

Motor imagery training (n = 10),

Physical practice (n = 10) and No practice (n = 10).

Golf putting performance.

Results showed that the accuracy of the putting performance were improved over time through the two types of cognitive training (AO and MI training).

Gonzalez-Rosa et al 2014

HS (n = 30), non-athletes, right-handed volunteers (17 females, 13 males, mean age

22.9 + 2.3 years).

Three groups:

AO watched a video of the task (n = 9), MI had to imagine it (n = 12), and CG with a distracting computation task (n = 9).

Early learning of a complex four limb, hand-foot coordination task, and kinematic analysis.

AO showed better learning compared with MI, and also elicited a stronger activity of the sensorimotor cortex during training, resulting in a lower amount of cortical activation during task execution.

During AO, subjects appear to process and collect sensory and motor information relevant to action in an effective and efficient manner, which allowed them to apply a series of decision making strategies appropriate to defining which movement sequence to perform, and activating control processes such as feed forward control during motor execution.

Hidalgo-Perez et al. 2015

HS (n = 40) 24 men and 16 women aged from 18 to 65 years.

Group 1: MI plus motor control exercise (n = 20),

Group 2: motor control exercise in isolation (n = 20).

Sensorimotor function of the craniocervical region and the cervical kinesthetic sense.

Combining MI with the motor control exercise produced statistically significant changes in sensorimotor function variables of the craniocervical region. Cervical kinesthetic sense was not significantly different between both groups.

Ingram et al. 2016

HS (n = 102)

4 groups:

MI or PP tested in either perceptual (altering the sensory cue) or motor (switching

the hand) transfer conditions (n = 60).

CG (n = 42) that did not perform a transfer condition.

Perceptual and motor learning through reaction time.

Results suggested that MI-based training relies on both perceptual and motor learning, while PP-based training relied more on motor processes.

Nishizawa & Kimura, 2017

HS females (n = 45) mean age

20.4 + 1.7 years).

Three groups:

Model- and self-observation (n = 15), model-observation (n = 15), and self-observation (n = 15).

Motor gesture learning through the acquisition of correct sports movement.

Observation combining model and self-observation exerted a positive effect on short-term

motor gesture learning.

Kawasaki et al 2018

Elderly HS (n = 36) aged

60 years or older (7 women and 29 men, mean age = 70.5 ± 6.19 years).

Three groups:

Unskilled or skilled model observation groups (n = 12, respectively), or the CG (n = 12).

Ball rotation performance (ball rotation speed).

Results indicated that the time taken for early phase learning of a finger coordination skill was improved when an unskilled model, rather than a skilled model, was used for AO combined with MI training.

Kraeutner et al. 2016

HS (n = 64) Right-handed participants (42 female, 22.1 ± 5.3 years).

Two groups:

MI in isolation (n = 31)

Physical practice (n = 33)

Implicit sequence learning task.

The magnitude of the learning did not differ between groups. It is suggested that MI and physical practice are equally effective in facilitating implicit sequence learning.

Kraeutner et al. 2017

HS (n = 72) Right-handed subjects (49 females, 23.8 ± 7.2 years)

Four conditions of MI-based practice:

4 training blocks with a high (4-High) or low (4-Low) sequence to noise ratio, or 2 training blocks with a high (2-High) or low (2 Low) sequence to noise ratio.

Implicit sequence learning task.

Results showed that the extent to which implicit sequence learning occurs through MI is impacted by manipulations to entire training time and the sequence to noise ratio. In addition, Results showed that the extent of implicit sequence learning occurring through MI is a function of exposure, indicating that like physical practice, the cognitive mechanisms of MI-based implicit sequence learning rely on the formation of stimulus response associations.

Lagravinese et al. 2016

HS (n = 25)

(AO) training: Subjects were exposed to the observation of a video showing finger tapping movements executed at 3Hz, a frequency higher than the spontaneous one (2Hz) for four consecutive days.

The changes in motor performance and motor resonance.

Results showed that multiple sessions of AO training induced a shift of the speed of execution of finger tapping movements toward the observed one and a change in motor resonance.

Lei et al. 2016

HS (n = 47) Right-handed individuals (23 men, 17 women), aged from 18 to 30 years.

Five conditions:

-AO in which the subjects watched a video of a model who adapted to a novel visuomotor rotation

-Proprioceptive training, in which the subject’s arm was moved passively to target locations that were associated with desired trajectories

-Combined training, in which the subjects watched the video

of a model during a half of the session and experienced passive

movements during the other half -Active training, in

which the subjects adapted actively to the rotation

-A control condition, in which the subjects did not perform any task.

Results showed an improvement in visuomotor adaptation following the action observation, as compared with the adaptation performed by the individuals who were naïve to the given visuomotor rotation

Salfi et al. 2019

HS (n = 39)

(Aged 24.9 ± 3.0 years; range, 20–34; 18 males).

MI and

Targeted memory reactivation.

Four conditions:

-MI in isolation

-MI with an incompatible sound stimulation

-AO

-Auditory targeted memory reactivation during AO

To assess the influence on performance on a sequential finger tapping task of an auditory targeted memory reactivation during MI practice.

The combination of MI and targeted memory reactivation showed the largest early performance improvement as indexed by the combined measure of speed and accuracy

Sobierajewicz et al. 2016

HS (n = 24)

6 males and 18 females range 21 to 28 years.

After an informative cue, a response sequence had either to be executed, imagined, or withheld.

The learning of a fine hand motor gesture.

Both physical condition and MI condition improved the response time and accuracy although the effect of motor learning by motor imagery was smaller than the effect of physical practice

EG: Experimental group, CG: Control group, MI: Motor Imagery; AO: Action observation; HS: Healthy subjects

Table 2. Assessment of the studies quality based on PEDro Scale

Items

1

2

3

4

5

6

7

8

9

10

11

Total

Cabral-Sequeira et al. 2016

1

1

0

1

0

0

0

1

0

1

1

5

Kumar et al. 2016

1

1

1

1

0

0

1

1

0

1

1

7

Keynen et al. 2018

1

1

1

1

0

0

0

1

0

0

1

5

La Touche et al. 2019

1

1

1

1

1

0

1

1

0

1

1

8

Moukarzel et al. 2019

1

1

0

1

1

0

1

1

0

1

1

7

Cuenca-Martínez et al. 2019

1

1

1

1

1

0

1

1

0

1

1

8

Bek et al. 2016

1

1

0

1

0

0

0

1

0

1

1

5

Sheahan et al. 2018

1

0

0

1

0

0

0

1

0

1

1

4

Dana & Gozalzadeh 2017

1

0

0

0

0

0

0

1

0

1

1

3

Kim et al. 2017

1

1

0

1

0

0

0

1

0

1

1

5

Gonzalez-Rosa et al. 2014

1

1

0

1

0

0

0

1

0

1

1

5

Hidalgo-Pérez et al. 2015

1

1

1

1

0

0

1

1

0

1

1

7

Ingram et al. 2016

1

1

0

1

0

0

0

1

0

1

1

5

Nishizawa & Kimura, 2017

1

1

0

1

0

0

0

1

0

1

0

4

Kawasaki et al. 2018

1

1

1

1

0

0

0

1

0

1

1

6

Kraeutner et al. 2016

1

1

0

0

0

0

0

0

0

1

1

4

Kraeutner et al. 2017

1

1

0

1

0

0

0

0

0

1

1

4

Lagravinese et al. 2016

1

0

0

1

1

0

0

1

0

1

1

5

Lei et al. 2016

1

0

0

1

0

0

0

1

0

1

0

3

Salfi et al. 2019

1

0

0

1

0

0

0

1

0

1

1

4

Sobierajewicz et al. 2016

1

0

0

1

0

0

0

0

0

1

1

3

1: subject choice criteria are specified; 2: random assignment of subjects to groups; 3: hidden assignment; 4: groups were similar at baseline; 5: all subjects were blinded; 6: all therapists were blinded; 7: all evaluators were blinded; 8: measures of at least one of the key outcomes were obtained from more than 85% of baseline subjects; 9: intention-to-treat analysis was performed; 10: results from statistical comparisons between groups were reported for at least one key outcome; 11: the study provides point and variability measures for at least one key outcome.

The English writing style of the present manuscript is very well accomplished, even so the manuscript is quite hard to follow.

Response: Thank you for this comment. English was edited and corrected by a company dedicated exclusively to it.

First, terms are not introduced or not consistently used. I will give two examples, but there are much inconsistently used terms. What is the difference between brain training (line 42) and movement representation techniques (title)? What is the difference between motor skills (line 42) and motor gestures (line 43)? If you intend to express the same thing, please be concise and use the same term throughout the whole manuscript. Second, the aim of the manuscript is not clear. In my opinion the aim of the manuscript cannot be to formulate a hypothesis, but rather to test a hypothesis. A theoretical framework is provided, but it remains unclear how this framework has been developed and where it comes from. Further, it remains unclear what this framework exactly explains. Is it the idea to make up a framework that integrates both MI and AO together?

Response: First, thank you for this comment. We agree that we must unify terms. You are absolutely right, and we have done so. Secondly, if you read any hypothesis study, you will see that this type of study has an idiosyncrasy of its own. I mean, first I slightly contextualize the field of study, then the hypothesis is formulated, and finally, it is tested with the available literature (in the case it can be tested, that is why it is a hypothesis). I think we should understand this as a different and unusual kind of study that we are all used to review for different prestigious international journals.

With respect to the theoretical framework, of course we have tried to create a framework that integrates both techniques, so the Table 1 is for both. But of course, although they share a large number of neurophysiological processes, these are not the same. That is why we have the Figure 1 and 2 to launch inferences about how vulnerable both tools are to the influence of different variables and how they could work in the process of creating a motor imprint as a prerequisite for learning. The text is thick because it incorporates a large number of concepts but we believe that it is understood that the two objectives of this hypothesis is to see how both techniques work (that is why the title is for both techniques) and also, in order to be able to differentiate the process of creating the mental image.

-The terms have been unified.

Many passages lack literature references. These are at least partially provided at later stages of the manuscript. However, they need to be included in the second part of the manuscript. I’d suggest to combine ‘2. Hypothesis’ and ‘3. Evidence’ to one part called ‘Theoretical framework’. It would add incremental value to the manuscript if the framework would be tested with data. If the framework is not tested in the present study, it should be stated how this could be done in future studies.

Response: Thank you very much for this comment but we cannot agree with it. The canon to follow is widely visible in the hypotheses that are published. After the context, you have to

formulate each of the hypotheses and then provide literature to support it or not. I insist, you can look at any hypothesis of specialized journals in hypotheses and you will see that what I say to you is like that and we believe that it is the most logical.

Even so, as we know that the reviewer's intention is to improve the present manuscript, we have replaced point 3 with the term "theoretical framework".

(see manuscript)

Again use of terms, but now the main terms. Please clearly distinguish between motor imagery and mental practice. The same holds for AO and using AO to improve motor execution. In parts of the manuscript, it remains unclear whether MI and ME differ in durations, differ in vividness, or whether the repeated use of MI and ME does not result in the same performance enhancements. Also explain what you mean with ‘movement representation training’ (line 33). It could mean learning a physical skill with movement representation techniques, but also to train to improve movement representation (more vivid in MI, better focus in AO). Further, I would recommend not to use the term ‘brain training’ (line 41 and others). I associate this with doing Sudoku grids to stay clever, but that could be my personal opinion.

Response: Thanks for the suggestions. We have normalized the terms and removed "brain training" from the manuscript.

Movement representation training means performing a task of motor imagery or action observation, I mean, mentally performing a motor gesture. But we don't use the term mental, we don't like the word “mental”. That's why we talk about training in movement representations. To represent a movement already carries implicitly the word “mental”, without needing to use it. To represent a movement implies to create it in the "mind", it is a way to unify the way to do it. I don't care if I create it with imagery or if they provide it with a video (action observation), what I want is to represent it in my brain without doing it in a real way.

-The terms have been unified.

In line 40 literature is cited that investigated MI in isolation, AO in isolation, both in combination with ‘various brain training modalities’, and in combination with real practice. I had the impression that you missed a whole bunch of studies of David Wright and colleagues who had been working on what they call AOMI. There has been very much work on this topic in the last five years. I would further suggest to include theory on MI and AO provided by Grush (2004) and Glover & Baran (2017). Then highlight more clearly which are the new insights of the present manuscript.

Response: Thank you for this contribution. These research groups are well known to us, believe me. If you look, this part of the manuscript is a little introduction, a bit of context simply. Obviously, there is a lack of studies, not only those that you tell me, but more (ours even) but I must insist on this: this work is a different type of study that has to be evaluated as such. Even so, we have slightly integrated these two studies you tell us.

(see main document)

“In this regard, Grush in 2004 proposed one of the most relevant theories in this field, the emulation theory of representation. This theory tries to establish a theoretical framework in which, during IM, the brain constructs a visual model between the body and the environment. Subsequently, these models produce or direct an efferent sensorimotor copy in order to provide expectations or predictions of sensory feedback. These models can also be run later to create new motor images, predict results of different actions or build new motor plans. This is the reason why visual perception is the result of using this type of models to create expectations and interpret sensory contributions during MI. In this sense, AO could provide that visual input between the subject's body and the environment, which could facilitate the process of constructing the mental image [10].

On the other hand, Glover & Baran have developed the Motor-Cognitive model of the MI. This model argues that central executive functions play a fundamental role during IM, but not so much in open actions. In this model, it is shown that the creation of motor mental images involves both a planning phase and a movement execution phase. To begin the creation of the mental image for the preparation of movement, an initial mental image is generated based on the motor representations stored in the nervous system. During MI and real execution, neurologically the processes are very similar, but nevertheless, during the execution of the mental task and the execution of the real task the processes change remarkably. During real movement, the nervous system unconsciously accesses processes of visual and proprioceptive feedback to refine the movement simultaneously with its execution. However, during MI, the control of movement creation is consciously dependent on the initial image created. That is why the ability to create motor mental images depends on the fidelity in which the subject can create the initial image. The widely developed motor actions are going to suppose a lower cognitive demand and a greater reliability in the representation, and on the contrary, the poor developed actions could to create an unreliable and unprecise motor images [11].”

In the beginning of page 2, motor learning by physical practice is explained. I believe this could be done with more detail. ‘The presentation of a novel gesture’ (line 47) only applies for motor learning by external instruction. However, I believe that that much of our motor skills repertoire is learned by exploration. This means doing something random and then remembering it to repeat it at later stages. This is how children and adults learn a new motor skill (not a new combination of already known skills). For instance, a new motor skill could be to move your ears. I am totally in line with the authors that oral instructions, AO, or MI may fasten the learning process. However, feedback (for moving the ears visual feedback from a mirror) should also be taken into account. Further, errors as a driving mechanism in motor learning should be taken into account. There are studies investigating action consequences in MI (Dahm & Rieger, 2019). There are also studies investigating the impact of action errors on motor learning in AO by manipulating the Video material (showing actors who make errors or not, showing real participants who naturally err).

Response: Thank you very much for this contribution, we agree with your words. You know that there are many different theoretical models to explain the process of acquiring new motor gestures. There are theories that travel from strict behaviourism to purely cognitive models (and mixed as well). Depending on the model you use, there will be a set of variables that you will have to use. Adams, Schmidt, Welford, Marteniuk, Fitts and Posner, Berstein, Gentile, ecological theory, etc. each follow their own paradigm. We try to choose a model widely accepted by all and make it simple because it is a small context. However, we believe that this contribution of yours (it is your work) can be integrated here and contribute to making this manuscript better.

With regard to verbal instructions, the state of the art supports that depending on the quality and quantity of the instructions, they may have the ability to facilitate or, on the contrary, hinder the overall learning process. The influence lies, above all, in the functions of focusing on the objectives and in the fact of being able to create coping strategies for a given challenge. Verbal instructions are largely related to another cognitive variable: the understanding of these. In the same way that in a process of constructing an image, that is, in a process of representation through an exercise in motor imagery, the understanding of instructions is decisive for the creation and construction of the image, for the real realisation of a motor gesture, simple, direct and easy to understand verbal instructions are also required.

The impact of verbal instructions is determining, above all, in those phases of motor learning where a high cognitive load predominates, and, therefore, in the first phases of the same. Here, the presence or not of a feedback can enhance the effect of these. Feedback can be defined, as Cano-de-la-Cuerda et al. argue, as information emanating from the results of motor actions, that is, from motor outputs. At least two types of feedback can be distinguished: an intrinsic type of feedback and an extrinsic type of feedback.

The first refers to the information that emerges after the results of the motor action and that is captured internally, that is, this information is incorporated thanks to the activity of the proprioceptive pathways, information related to the knowledge of the position of the body, and exteroceptive, by the action of the cutaneous receptors, sight, hearing, etc. which allow the correction and adjustment of movements. On the other hand, the second feedback, called extrinsic, is that information that emanates from a place or source external to the body of the person. Both forms of feedback form a complex that influences the progression and improvement towards learning a certain motor task.

External feedback can be classified into two forms: knowledge of the achievement or knowledge of the outcome. Therefore, the external source can provide verbal information to the subjects referring to the motor patterns of the execution of a movement in question or, with respect to the result, especially interesting when the internal feedback is limited for any reason whatsoever. Some research has been carried out to answer the question of which of the two forms of external feedback is best.

Biloudeau et al. found that knowledge of results is a very important learning variable to take into account in the process of acquiring new motor tasks. However, there are certain types of motor tasks for which intrinsic feedback, such as visual or kinesthetic, is sufficient to provide most of the information concerning errors, and knowledge of performance has only minimal effects. For example, in the learning of follow-up tasks, knowledge of results only minimally improves the performance and learning of a movement.

Cano-de-la-Cuerda and its collaborators further argue that external feedback can be used to obtain verbal information on progress in the motor learning process, in the classification and schematization of the entire movement in order to be divided into smaller, simpler movements in the movement integration process, it can also play a positive reinforcing role by improving not only adherence, but also by increasing motivation levels which is a key variable in the learning process, and so on. It seems to be widely important to bear in mind that people should receive information from both internal and external sources in order to improve the motor learning process.

To finish, this is what we added to the text:

- “In addition to this, it is important to stress that the repertory of motor gestures can be learned through an exploratory process. Above all, novel motor gestures. The feedback mechanism can help the learning process, as for example having knowledge of mistakes can consolidate the improved acquisition of a given motor gesture. Several authors have investigated the importance of feedback in motor learning process [12–14].”

References

Cano-de-la-Cuerda, R.; Molero-Sánchez, A.; Carratalá-Tejada, M.; Alguacil-Diego, I.M.; Molina-Rueda, F.; Miangolarra-Page, J.C.; Torricelli, D. Teorías y modelos de control y aprendizaje motor. Aplicaciones clínicas en neurorrehabilitación. Neurologia. 2015, 30, 32–41. Bilodeau, E.A.; Bilodeau, I.M.; Schumsky, D.A. Some effects of introducing and withdrawing knowledge of results early and late in practice. J. Exp. Psychol. 1959, 58, 142–144. Dahm, S.F.; Rieger, M. Is imagery better than reality? Performance in imagined dart throwing. Hum. Mov. Sci. 2019, 66, 38–52.

Please explain where the four categories (physical, cognitive, motivational-emotional, direct modulation) come from. Provide references and explain how these categories were selected. The last category seems not to fit to the other three categories. Further, explain why you’d expect the influences described in Table 1 and provide references. Where do the starts in Table 1 come from? Please explain and provide references. I like the idea to disentangle all the variables listed in Table 1. However, this could be done more concisely and not necessarily in a table. What do you mean with ‘influence’? 1. The influence of the variable on the ease/vividness of MI? Is there something similar to ease/vividness in AO? 2. The influence of the variable on motor learning (improvements in physical execution of the action). Maybe explain the four possible types of influence for each variable separately?

Response: Thank you for these suggestions. First, we very much agree with you that a paragraph is needed to support studies because the authors of this manuscript believe that these variables will influence the process of representation of a motor image. This paragraph has been added and we thank the reviewer for this request.

However, it seems logical that a task of constructing a mental image is influenced by cognitive variables, emotional variables, motivation or the ability to integrate somatosensory information. Who would be willing to deny this? It would be like going back several years.

The group of variables of direct modulation is effectively in another different dimension. However it is necessary because it is not the same to create a mental image than to watch a video, therefore, there are a series of variables at the time of doing so that are going to have a very important impact "I insist on the time to do it". This group satisfies this dimension.

This can easily answer the question you ask me: What does it mean to influence? Well, influencing means "ability to modulate the effect". If I do not understand the verbal instructions, or the gesture is too complex, I am going to have more difficulty creating a mental image than if I am shown a video. I think it answers your question.

It is explained in text as well as in the table. In the table we did it because it is more visual (as a summary, we believe that the reader can better understand it this way, do not you think?). When you suggest that we separate the influence on each tool I don't see it right and excuse me for not agreeing. I argue the reason: The influence, I mean, the reason is the same, it's the same for both of us (is totally shared). But it is the technique itself that exhibits more or less susceptibility to that influence. Finally, with respect to the susceptibility or influence on each motion representation tool is just a hypothesis. You will not find it in any study. This is an attempt to explain the results of various studies.

(Hypothesis section): “Several studies support the presence of these variables related to movement representation techniques. For example, regarding the cognitive variables, greater mental efforts made during an imagery tasks led to greater hemodynamic changes at cortical level [15]. Regarding the physical domain, there is extensive literature that supports their influence on the process of movement representation. For example, athletes with high levels of physical activity had a greater ability to generate motor images than amateur athletes with less levels of physical activity [16–18]. The study conducted by La Touche et al. 2018[19] showed that patients with chronic low back pain presented a negative correlation between the level of kinesiophobia and the ability to generate both kinesthetic and visual motor images. In addition, they also found that the ability to generate motor images was impaired in patients with chronic low-back pain compared with healthy participants. This also was found by another research group [20].

With respect the direct modulation variables, providing visual input before to performing an imagery motor task facilitates it and causes greater neurophysiological activity than if performed alone [21–23]. In addition, it has been found that the vividness of the imagination affected motor learning, showing more significant changes in those participants who presented a more vivid imagination [24]. Regarding the autonomic nervous system response, Cuenca-Martínez et al. found that the complexity of movement, the effort-intensity and the levels of physical activity can influence neurovegetative activity in the process of generating motor images [25]. Finally, regarding the synchronization, several studies have showed that unknown, uncommon and uncomfortable movements can lead to differences between the time employed between the imagined and real execution [26,27].”

References

Wriessnegger, S.C.; Kirchmeyr, D.; Bauernfeind, G.; Müller-Putz, G.R. Force related hemodynamic responses during execution and imagery of a hand grip task: A functional near infrared spectroscopy study. Brain Cogn. 2017, 117, 108–116. Williams, S.E.; Guillot, A.; Di Rienzo, F.; Cumming, J. Comparing self-report and mental chronometry measures of motor imagery ability. Eur. J. Sport Sci. 2015, 15, 703–711. Di Corrado, D.; Guarnera, M.; Quartiroli, A. Vividness and Transformation of Mental Images in Karate and Ballet. Percept. Mot. Skills 2014, 119, 764–773. Paris-Alemany, A.; La Touche, R.; Agudo-Carmona, D.; Fernández-Carnero, J.; Gadea-Mateos, L.; Suso-Martí, L.; Cuenca-Martínez, F. Visual motor imagery predominance in professional Spanish dancers. Somatosens. Mot. Res. 2019, 1–10. La Touche, R.; Grande-Alonso, M.; Cuenca-Martínez, F.; Gónzález-Ferrero, L.; Suso-Martí, L.; Paris-Alemany, A. Diminished Kinesthetic and Visual Motor Imagery Ability in Adults With Chronic Low Back Pain. PM&R 2018. Pijnenburg, M.; Brumagne, S.; Caeyenberghs, K.; Janssens, L.; Goossens, N.; Marinazzo, D.; Swinnen, S.P.; Claeys, K.; Siugzdaite, R. Resting-State Functional Connectivity of the Sensorimotor and the Association with the Sit-to-Stand-to-Sit Task. Brain Connect. 2015, 5, 303–11. Taube, W.; Mouthon, M.; Leukel, C.; Hoogewoud, H.-M.; Annoni, J.-M.; Keller, M. Brain activity during observation and motor imagery of different balance tasks: An fMRI study. Cortex 2015, 64, 102–114. Vogt, S.; Rienzo, F. Di; Collet, C.; Collins, A.; Guillot, A. Multiple roles of motor imagery during action observation. Front. Hum. Neurosci. 2013, 7, 807. Sakamoto, M.; Muraoka, T.; Mizuguchi, N.; Kanosue, K. Combining observation and imagery of an action enhances human corticospinal excitability. Neurosci. Res. 2009, 65, 23–27. Isaac, A.R.; Marks, D.F. Individual differences in mental imagery experience: developmental changes and specialization. Br. J. Psychol. 1994, 85, 479–500. Cuenca-Martínez, F.; Suso-Martí, L.; Grande-Alonso, M.; Paris-Alemany, A.; La Touche, R. Combining motor imagery with action observation training does not lead to a greater autonomic nervous system response than motor imagery alone during simple and functional movements: a randomized controlled trial. PeerJ 2018, 6, e5142. Parsons, L.M. Temporal and kinematic properties of motor behavior reflected in mentally simulated action. J. Exp. Psychol. Hum. Percept. Perform. 1994, 20, 709–30. Rieger, M. Motor imagery in typing: effects of typing style and action familiarity. Psychon. Bull. Rev. 2012, 19, 101–107.

The three paragraphs (line 192-228) are 3 independent summaries of three previous studies. A red line is missing as in many parts of the manuscript. There are many interesting points raised in the manuscript, but they are not logically connected. They are rather set one after another. The same holds for the information in the figures.

Minor Comments

I would suggest to structure the sentences in a way that the authors are not mentioned in the text. This would not only shorten the manuscript, but also make it easier to read. I will give an example: ‘MI is defined as a cognitive and dynamic ability involving the cerebral representation of an action without its real motor execution [3]’ (line 36).

Response: Thank you very much for this suggestion, we have changed most (although not all due to the importance of the author)

- MI is defined as a cognitive and dynamic ability involving the cerebral representation of an action, without its real motor execution[3]. AO training is considered as the internal representation of a set of movements evoked by the observer during live visualization of the movements[4].

- In addition to this, it is important to stress that the repertory of motor gestures can be learned through an exploratory process. Above all, novel motor gestures. The feedback mechanism can help the learning process, as for example having knowledge of mistakes can consolidate the improved acquisition of a given motor gesture. Several authors have investigated the importance of feedback in motor learning process [12–14].

- However, it has been reported that this activation is lower during movement representation than during actual practice [32]

- It has been found that the amplitude of the evoked motor potentials during AO and MI correlated positively with the ability to generate motor images [40].

- However, It has been reported that both explicit and implicit motor learning processes can occur [43].

- The information is consciously retained in the working memory for subsequent processing to guide behaviors [47]

- Several studies have found that AO training led to greater motor learning of complex gestures in the short term than did MI [56,57].

Abstract line 27: ‘AO is less demanding in terms of cognitive load than MI’.

Response: Thank you for this adjustment.

- AO is less demanding in terms of cognitive load than MI

Please use ‘physical practice’ instead of ‘actual practice’. It fits better in contrast to mental practice.

Response: Thank you for this suggest

(see main text)

Line 77: ‘the most susceptible of the movement representation techniques’. How many techniques are there? So far you present only two: MI and AO. Did I miss something?

Response: Thank you for the detail. It's a writing error.

Our hypothesis is that MI is more susceptible than AO to the influence of these key variables, due to the inherent characteristics of the motor image construction process.

Table 1 ‘physical activity level’: Do you mean that it is easier imagining a high jump while riding a bike?

Response: I mean the amount of METs (how much physical activity do you do).

Table 1 ‘fatigue’: perceived fatigue and objective muscle fatigue is not the same. Please be precise in the description of the influence.

Response: Thank you for the clarification. I am referring to mental fatigue.

- Perceived of mental fatigue

Figure 1: What is the new insight that I get from this framework? Is it the four categories in black? Please explain this to the reader. Also, there are quite a lot of arrows in the figure. Please explain them in the text and provide references with data that made you conclude this.

Response: Thank you for this suggestion. In the text it was slightly explained but we have expanded it. We thought it was understood without any problem. With respect to the references, they are the same variables as the table (although they are sorted by domains), but they have already been included previously. To include it here again would be redundant. In addition, I mean, I understand what you want to say but look, the presence of the variables is justifiable with studies, but how they act is part of the hypothesis. It seems to me that it makes little sense to reference this hypothesis.

We also propose a categorization system related to the influence of these variables on the process of movement representation. The primary variables are the direct modulation factors because they act directly on the process of live movement representation. Cognitive and physical variables could influence the direct modulation variables and the motor learning process. For example, physical activity levels could increase to generate more experience and thereby facilitate the generation of motor images. This process would also improve the understanding of the motor gesture, thereby facilitating the ability to perform the mental representation of movement. Motivational-emotional variables could influence all of these variables at all steps in the process. The visual information can help the creation of the motor representation and the set of direct modulation variables as it can facilitate this process. This has been demonstrated in multiple studies [15–18]. The creation of the motor representation provokes a neurophysiological activation qualitatively similar to that which occurs during physical practice. This has even been shown with neurovegetative activity [19]. The result of this process is the generation of mnemonic representations of movements as a prerequisite to motor learning. Figure 1 graphically represents this categorization system.

Line 123: Is tools and techniques the same? Is there a difference?

Response: Yes, they are the same things.

Line 124 ‘different strategies’: Please explain them in detail.

Response: This is part of the hypothesis. All the text that follows is to explain this. Different strategies are referred to "in global" and then detailed in what is different, I cannot understand this comment

“The first of these arguments is that the neurophysiological paths followed by the two movement representation tools (AO and MI) during the process of acquiring and integrating visual information differ. Therefore, different strategies are employed in the process of creating the motor print. The first argument introduces the second.

The second argument is that image construction through MI is likely fed initially by the continuous activity of the working memory and then through the activity of the episodic buffer. Figure 2 shows how this operative memory activity acts in order to integrate the visual information feeding the image construction. However, Figure 2 also shows that image construction will also receive information from episodic memory. Episodic memory feeds and is fed by semantic memory and, in the same way, by perceptual memory. Therefore, MI requires predominantly conscious strategies for the image creation process and thus a high cognitive load, which could explain the fatigue experienced during the image construction process through MI. However, it is important to stress that it is also possible to generate images relatively unconscious on some occasions such as during reading. But, predominantly MI needs conscious strategies.

The third argument is that AO is not necessarily dependent on the use of conscious strategies due to the efficiency of externally provided images. In AO, the main task is to retain and understand the image rather than creating it, facilitating the working memory tasks and thus the construction of the motor print. As a result, image transformation and conscious effort can occur during AO but likely requires less effort than for MI.

The fourth argument is that this neurophysiological activity is optimized between the central executive control (which is part of the working memory) and procedural memory, thereby enabling the acquisition of strategies while being unaware of the processes that govern the acquisition of those strategies. Thus, during the process of creating the motor print through AO, there is likely to be greater involvement of implicit learning with the participation of the perceptive-motor procedural memory.

The fifth and last argument is that this activity could also respond to differences between AO and MI in susceptibility to the influence of physical, cognitive, motivational-emotional and direct modulation variables, showing greater robustness for the influence of AO training (Figure 2).”

On page 6 it is explained that AO is unconscious, but MI is conscious. This may not always be the case. For instance, if someone reads a text that includes action words, the reader may imagine the action involuntarily.

Response: Thank you very much for this contribution, this is really as you say. This has been modified

“Therefore, MI requires predominantly conscious strategies for the image creation process and thus a high cognitive load, which could explain the fatigue experienced during the image construction process through MI. However, it is important to stress that it is also possible to generate images relatively unconscious on some occasions such as during reading. But, predominantly MI needs conscious strategies.”

Line 145: ‘The difference between MI AO is that all participants have the same afferent visual information arriving for processing in AO …’. I agree that the video should be the same for all subjects. But similarly the instructions in MI should always be the same for all subjects. This is the objective stimulus material which is the same for all subjects in both MI and AO. What participants do with the stimulus material (instructions, videos etc.) is what is here called ‘processing’. I would say that both in MI and AO, participants may have an individual focus on certain details of the action. Please explain why this is different in MI and AO.

Response: Thank you very much for this excellent contribution, this is also really as you say. This has been slightly modified.

The difference between MI and AO is that all participants have the same afferent visual information arriving for processing in AO, while in MI, even though everyone receives the same verbal instructions, it is likely that there are likely to be interindividual variations that could modulate the potential of MI and consequently the effect of MI on learning. The success of MI depends mainly on each individual’s ability to create motor images. It will also depend on the set of variables previously mentioned with the system of integration of somatosensory information, motivation, levels of physical activity, among others.

Line 149: ‘…efficiency of the mirror neuron system is greater during AO…’ than during MI. Please further explain what you mean with ‘efficiency’. Do the neurons fire less often with a higher outcome? What is the outcome? Please provide references for your statement.

Response: It implies that the functional connection visual-premotor cortex needs less resources to achieve the same firing rate. I recommend that you read Dr. Gatti's research.

This has been explicitly reported by Gatti et al.[31].

Gatti, R.; Tettamanti, A.; Gough, P.M.; Riboldi, E.; Marinoni, L.; Buccino, G. Action observation versus motor imagery in learning a complex motor task: A short review of literature and a kinematics study. Neurosci. Lett. 2013, 540, 37–42.

Line 168: ‘the two practices’ which ones do you mean? MI? AO? Physical practice?

[…] Between the physical and non-physical practice […]

I am not sure whether figure 2 improves readability and understanding of the text. Particularly because the content is not further explained in the text.

Response: Thank you very much for the suggestion. We have added information regarding this request

“The first of these arguments is that the neurophysiological paths followed by the two movement representation tools (AO and MI) during the process of acquiring and integrating visual information differ. Therefore, different strategies are employed in the process of creating the motor print. The first argument introduces the second.

The second argument is that image construction through MI is likely fed initially by the continuous activity of the working memory and then through the activity of the episodic buffer. Figure 2 shows how this operative memory activity acts in order to integrate the visual information feeding the image construction. However, Figure 2 also shows that image construction will also receive information from episodic memory. Episodic memory feeds and is fed by semantic memory and, in the same way, by perceptual memory. Therefore, MI requires predominantly conscious strategies for the image creation process and thus a high cognitive load, which could explain the fatigue experienced during the image construction process through MI. However, it is important to stress that it is also possible to generate images relatively unconscious on some occasions such as during reading. But, predominantly MI needs conscious strategies.

The third argument is that AO is not necessarily dependent on the use of conscious strategies due to the efficiency of externally provided images. In AO, the main task is to retain and understand the image rather than creating it, facilitating the working memory tasks and thus the construction of the motor print. As a result, image transformation and conscious effort can occur during AO but likely requires less effort than for MI.

The fourth argument is that this neurophysiological activity is optimized between the central executive control (which is part of the working memory) and procedural memory, thereby enabling the acquisition of strategies while being unaware of the processes that govern the acquisition of those strategies. Thus, during the process of creating the motor print through AO, there is likely to be greater involvement of implicit learning with the participation of the perceptive-motor procedural memory.

The fifth and last argument is that this activity could also respond to differences between AO and MI in susceptibility to the influence of physical, cognitive, motivational-emotional and direct modulation variables, showing greater robustness for the influence of AO training (Figure 2).”

Line 279: What do the presented results have to do with ‘improvements in the mirror neuron system’?

Response: I think you're right here. I was really referring to the efficiency of its function. We have eliminated it. Thank you very much for this contribution.

Reference 8 (line 334). ‘Le Effects of motor imagery’

Response: Corrected

Reviewer 2 Report

This is a very interesting and important pice of work. I highlight the strengths and weaknesses of the work below and then outline some recommendations for minor revisions. 

Strengths:

The authors have provided a comprehensive hypothesis about the role of action observation (AO) and motor imagery (MI) in motor learning. The strength of the current work lies in the discussion of variables, including their respective domains, and how it modulates the magnitude of brain activity during movement representation techniques. The provision of how AO and MI are susceptible to these variables would make a great contribution to the current knowledge governing the shared neurophysiological network of movement representation and execution.

Weaknesses:

While the authors have cited a substantial amount of evidence in most of the key aspects of the hypothesis, there are some areas that are left out and may need further clarification (or additional discussion and references) to improve the quality of the current work. For example, some variables discussed in the hypothesis have been mentioned but not clearly discussed in the evidence section. Having discussed these, the following recommendations have been made.

Recommendations:

“Perception of difficulty” in Table 1 (Page 4 of 15) – the authors have stated: “greater perception of the difficulty could lead to a reduced ability to generate motor representation and thereby worsen motor learning”. Please provide further clarification in the evidence section (Page 6 of 15 – section 3) about how this inference was made. The role of error and feedback might provide a strong support to the variables of modulation especially in the cognitive and direct modulation domains. If this is out of scope of the current work, please clarify in the introduction (section 1). Both the figures provide a vague flow of information to the reader. The authors should provide explanation about what certain arrows represent and how each function (especially in Page 8 of 15 - Figure 2) is connected in the network. Pagination issues – the count restarts after Page 7 of 15

Author Response

December 19, 2019

Dear Editorial Office,

We are pleased to submit our point-by-point response to the changes requested on our paper “The Role of Movement Representation Techniques in the Motor Learning Process: A Neurophysiological Hypothesis and a Systematic Review".

We believe that we have made all the changes requested prior to publication and our manuscript is now responsive to all comments and suggestions. We highlighted the changes along the text with red color to show reviewers and editor where the changes have been made. In addition, our responses to the reviewers in this document are highlighted in red color.

We would like to thank the journal Editor of Brain Sciences for having managed those different referring researchers in this area could have reviewed our manuscript. Thanks to them, the manuscript has remarkably improved its quality.

We look forward to hearing your response and thank you for your consideration in bringing this manuscript closer to publication.

Sincerely yours,

The authors

Reviewer 2

This is a very interesting and important pice of work. I highlight the strengths and weaknesses of the work below and then outline some recommendations for minor revisions.

Strengths:

The authors have provided a comprehensive hypothesis about the role of action observation (AO) and motor imagery (MI) in motor learning. The strength of the current work lies in the discussion of variables, including their respective domains, and how it modulates the magnitude of brain activity during movement representation techniques. The provision of how AO and MI are susceptible to these variables would make a great contribution to the current knowledge governing the shared neurophysiological network of movement representation and execution.

Response: Thank you so much for the words.

Weaknesses:

While the authors have cited a substantial amount of evidence in most of the key aspects of the hypothesis, there are some areas that are left out and may need further clarification (or additional discussion and references) to improve the quality of the current work. For example, some variables discussed in the hypothesis have been mentioned but not clearly discussed in the evidence section. Having discussed these, the following recommendations have been made.

Response: Thank you very much for all these comments. This is shared with the other reviewer. The authors of this manuscript have understood that we had to make a justification by means of some published research in order to be able to argue why these variables are important and not others. This has been added in the main document.  We are very grateful for these words.

Recommendations:

“Perception of difficulty” in Table 1 (Page 4 of 15) – the authors have stated: “greater perception of the difficulty could lead to a reduced ability to generate motor representation and thereby worsen motor learning”. Please provide further clarification in the evidence section (Page 6 of 15 – section 3) about how this inference was made.

Response: Thank you for this suggestion. The other reviewer asked not only to justify a variable. He asked to justify them all. If it seems appropriate to you, we have written a little text that justifies all the domains to make it more complete.

We also propose a categorization system related to the influence of these variables on the process of movement representation. The primary variables are the direct modulation factors because they act directly on the process of live movement representation.  Cognitive and physical variables could influence the direct modulation variables and the motor learning process. For example, physical activity levels could increase to generate more experience and thereby facilitate the generation of motor images. This process would also improve the understanding of the motor gesture, thereby facilitating the ability to perform the mental representation of movement. Motivational-emotional variables could influence all of these variables at all steps in the process. The visual information can help the creation of the motor representation and the set of direct modulation variables as it can facilitate this process. This has been demonstrated in multiple studies [15–18]. The creation of the motor representation provokes a neurophysiological activation qualitatively similar to that which occurs during physical practice. This has even been shown with neurovegetative activity [19]. The result of this process is the generation of mnemonic representations of movements as a prerequisite to motor learning. Figure 1 graphically represents this categorization system.

The role of error and feedback might provide a strong support to the variables of modulation especially in the cognitive and direct modulation domains. If this is out of scope of the current work, please clarify in the introduction (section 1).

Response: Thank you very much for this contribution, we agree with your words. The other reviewer also asked us to name the role of feedback very slightly. As specific researchers to the topic we would have a lot to say but we believe that the focus of the article would be somewhat lost. However, under the following text we show you the words we have added. Thank you very much for the suggestion.

There are many different theoretical models to explain the process of acquiring new motor gestures. There are theories that travel from strict behaviourism to purely cognitive models (and mixed as well). Depending on the model you use, there will be a set of variables that you will have to use.  Adams, Schmidt, Welford, Marteniuk, Fitts and Posner, Berstein, Gentile, ecological theory, etc. each follow their own paradigm. We try to choose a model widely accepted by all and make it simple because it is a small context.

With regard to verbal instructions, the state of the art supports that depending on the quality and quantity of the instructions, they may have the ability to facilitate or, on the contrary, hinder the overall learning process. The influence lies, above all, in the functions of focusing on the objectives and in the fact of being able to create coping strategies for a given challenge. Verbal instructions are largely related to another cognitive variable: the understanding of these. In the same way that in a process of constructing an image, that is, in a process of representation through an exercise in motor imagery, the understanding of instructions is decisive for the creation and construction of the image, for the real realisation of a motor gesture, simple, direct and easy to understand verbal instructions are also required.

The impact of verbal instructions is determining, above all, in those phases of motor learning where a high cognitive load predominates, and, therefore, in the first phases of the same. Here, the presence or not of a feedback can enhance the effect of these. Feedback can be defined, as Cano-de-la-Cuerda et al. argue, as information emanating from the results of motor actions, that is, from motor outputs. At least two types of feedback can be distinguished: an intrinsic type of feedback and an extrinsic type of feedback.

The first refers to the information that emerges after the results of the motor action and that is captured internally, that is, this information is incorporated thanks to the activity of the proprioceptive pathways, information related to the knowledge of the position of the body, and exteroceptive, by the action of the cutaneous receptors, sight, hearing, etc. which allow the correction and adjustment of movements. On the other hand, the second feedback, called extrinsic, is that information that emanates from a place or source external to the body of the person. Both forms of feedback form a complex that influences the progression and improvement towards learning a certain motor task.

External feedback can be classified into two forms: knowledge of the achievement or knowledge of the outcome. Therefore, the external source can provide verbal information to the subjects referring to the motor patterns of the execution of a movement in question or, with respect to the result, especially interesting when the internal feedback is limited for any reason whatsoever. Some research has been carried out to answer the question of which of the two forms of external feedback is best.

Biloudeau et al. found that knowledge of results is a very important learning variable to take into account in the process of acquiring new motor tasks. However, there are certain types of motor tasks for which intrinsic feedback, such as visual or kinesthetic, is sufficient to provide most of the information concerning errors, and knowledge of performance has only minimal effects. For example, in the learning of follow-up tasks, knowledge of results only minimally improves the performance and learning of a movement.

Cano-de-la-Cuerda and its collaborators further argue that external feedback can be used to obtain verbal information on progress in the motor learning process, in the classification and schematization of the entire movement in order to be divided into smaller, simpler movements in the movement integration process, it can also play a positive reinforcing role by improving not only adherence, but also by increasing motivation levels which is a key variable in the learning process, and so on. It seems to be widely important to bear in mind that people should receive information from both internal and external sources in order to improve the motor learning process.

To finish, this is what we added to the text:

- “In addition to this, it is important to stress that the repertory of motor gestures can be learned through an exploratory process. Above all, novel motor gestures. The feedback mechanism can help the learning process, as for example having knowledge of mistakes can consolidate the improved acquisition of a given motor gesture. Several authors have investigated the importance of feedback in motor learning process [12–14].”

Both the figures provide a vague flow of information to the reader. The authors should provide explanation about what certain arrows represent and how each function (especially in Page 8 of 15 - Figure 2) is connected in the network.

Response: Thank you very much for this suggestion. We fully agree with you. We have expanded the explanation of both figures so that it is better understood. We very much appreciate the reviewer's information. The authors think that he/she has helped make this manuscript better. Thank you very much.

Figure 1

“We also propose a categorization system related to the influence of these variables on the process of movement representation. The primary variables are the direct modulation factors because they act directly on the process of live movement representation.  Cognitive and physical variables could influence the direct modulation variables and the motor learning process. For example, physical activity levels could increase to generate more experience and thereby facilitate the generation of motor images. This process would also improve the understanding of the motor gesture, thereby facilitating the ability to perform the mental representation of movement. Motivational-emotional variables could influence all of these variables at all steps in the process. The visual information can help the creation of the motor representation and the set of direct modulation variables as it can facilitate this process. This has been demonstrated in multiple studies [15–18]. The creation of the motor representation provokes a neurophysiological activation qualitatively similar to that which occurs during physical practice. This has even been shown with neurovegetative activity [19]. The result of this process is the generation of mnemonic representations of movements as a prerequisite to motor learning. Figure 1 graphically represents this categorization system.”

Figure 2

“The first of these arguments is that the neurophysiological paths followed by the two movement representation tools (AO and MI) during the process of acquiring and integrating visual information differ. Therefore, different strategies are employed in the process of creating the motor print. The first argument introduces the second.

The second argument is that image construction through MI is likely fed initially by the continuous activity of the working memory and then through the activity of the episodic buffer. Figure 2 shows how this operative memory activity acts in order to integrate the visual information feeding the image construction. However, Figure 2 also shows that image construction will also receive information from episodic memory. Episodic memory feeds and is fed by semantic memory and, in the same way, by perceptual memory. Therefore, MI requires predominantly conscious strategies for the image creation process and thus a high cognitive load, which could explain the fatigue experienced during the image construction process through MI. However, it is important to stress that it is also possible to generate images relatively unconscious on some occasions such as during reading. But, predominantly MI needs conscious strategies.

The third argument is that AO is not necessarily dependent on the use of conscious strategies due to the efficiency of externally provided images. In AO, the main task is to retain and understand the image rather than creating it, facilitating the working memory tasks and thus the construction of the motor print. As a result, image transformation and conscious effort can occur during AO but likely requires less effort than for MI.

The fourth argument is that this neurophysiological activity is optimized between the central executive control (which is part of the working memory) and procedural memory, thereby enabling the acquisition of strategies while being unaware of the processes that govern the acquisition of those strategies. Thus, during the process of creating the motor print through AO, there is likely to be greater involvement of implicit learning with the participation of the perceptive-motor procedural memory.

The fifth and last argument is that this activity could also respond to differences between AO and MI in susceptibility to the influence of physical, cognitive, motivational-emotional and direct modulation variables, showing greater robustness for the influence of AO training (Figure 2).” 

Pagination issues – the count restarts after Page 7 of 15

Response: Thank you very much for seeing this detail. By changing the section to create a page in horizontal direction everything fell down. We tried to fix it but I think the editing team should help us because we don't know how to do it. We apologize.